# It Takes a Graph to Know a Graph: Rewiring for Homophily with a Reference Graph

## Abstract

Graph Neural Networks (GNNs) excel at analyzing graph-structured data but struggle on heterophilic graphs, where connected nodes often belong to different classes. While this challenge is commonly addressed with specialized GNN architectures, graph rewiring remains an underexplored strategy in this context. We provide theoretical foundations linking edge homophily, GNN embedding smoothness, and node classification performance, motivating the need to enhance homophily. Building on this insight, we introduce a rewiring framework that increases graph homophily using a *reference graph*, with theoretical guarantees on the homophily of the rewired graph. To broaden applicability, we propose a label-driven diffusion approach for constructing a homophilic reference graph from node features and training labels. Through extensive simulations, we analyze how the homophily of both the original and reference graphs influences the rewired graph homophily and downstream GNN performance. We evaluate our method on 13 real-world heterophilic datasets and show that it outperforms existing rewiring techniques and specialized GNNs for heterophilic graphs, achieving improved node classification accuracy while remaining efficient and scalable to large graphs.

## 1 Introduction

Graph-structured data naturally arises when modeling complex relationships between entities. Graph Neural Networks (GNNs) have gained prominence as an effective approach to analyzing such data, with applications spanning numerous domains (Li et al., 2021; Wu et al., 2020; Strokach et al., 2020). Most GNNs utilize a message passing mechanism that iteratively updates node representations by aggregating information from neighboring nodes, effectively leveraging both the graph structure and node attributes.

Standard GNNs are designed primarily for homophilic graphs, as they rely on the homophily assumption (i.e., "birds of a feather flock together") (McPherson et al., 2001), where connected nodes typically belong to the same class. When applied to heterophilic graphs, where neighboring nodes often belong to different classes, their performance deteriorates (Zheng et al., 2022; Zhu et al., 2020; Yan et al., 2022). In these cases, the message passing mechanism results in less distinguishable representations, ultimately reducing classification accuracy. To address this challenge, various GNN architectures have been designed to better handle heterophilic graph structures (Zhu et al., 2020; Abu-El-Haija et al., 2019; Chien et al., 2020; Chen et al., 2023).

Graph rewiring (Attali et al., 2024) is a method that decouples the original graph from the one used for message passing by modifying the graph's connectivity, thereby altering the flow of information and improving node classification performance. It is commonly employed to address two fundamental challenges in GNNs: over-smoothing (Li et al., 2018) and over-squashing (Alon & Yahav, 2020). Over-smoothing occurs when excessive message passing causes node representations to become indistinguishable. Conversely, over-squashing arises when too much information is compressed into a small subset of nodes, limiting the model's ability to capture long-range dependencies and, in practice, reducing its expressive power.

Despite the successful application of rewiring techniques to address over-smoothing and over-squashing in GNNs, their potential for enhancing GNN performance on heterophilic graphs remains largely unexplored. Specifically, leveraging graph rewiring to increase homophily—as an alternative

to designing specialized GNN architectures tailored for heterophilic data—represents a promising yet underdeveloped research direction. Here, we show that homophily-enhancing rewiring can significantly improve GNN performance in heterophilic settings, offering a complementary approach to existing architectural solutions.

In this paper, we have established both theoretical and empirical connections between edge homophily, the smoothness of learned GNN embeddings on the graph, and GNN performance. Our theoretical analysis shows that in graphs with low homophily, learned node embeddings that allow for an accurate classification cannot be smooth on the graph. This creates a conflict with the message passing mechanism of GNNs, which in many cases tends to smooth embeddings along the graph structure. This result provides strong motivation for improving graph homophily through rewiring. To this end, we propose a rewiring framework based on the concept of a *reference graph*, defined as a graph that shares the same node and label sets as the original graph but differs in its edge set. We further show that, under specific conditions related to the homophily of the reference graph relative to the original graph, rewiring the edge set using this framework *guarantees homophily enhancement*, and we support this theoretical result with illustrative simulations demonstrating improved performance following the homophily enhancement.

To implement our framework and systematically control graph homophily, we employ a label-driven diffusion approach (Mendelman & Talmon, 2025) originated in manifold learning (Coifman & Lafon, 2006) to construct a homophilic reference graph by leveraging underutilized information in this context – node features and training labels. When the resulting reference graph satisfies the conditions of our rewiring framework, it provably increases the homophily of the rewired graph. We validate our method through extensive experiments across a wide range of datasets and GNN architectures, consistently observing performance improvements. Our approach outperforms existing rewiring techniques and specialized GNNs for heterophilic graphs, effectively enhancing homophily and node classification accuracy.

## 2 RELATED WORK

**Graph homophily metrics.** Several types of graph homophily metrics have been proposed to capture different aspects of label correlation in graphs. These include edge homophily (Abu-El-Haija et al., 2019), node homophily (Pei et al., 2020), class homophily (Lim et al., 2021), neighbor homophily (Gong et al., 2023), and others. Each of these measures highlights a distinct aspect of homophily, depending on the focus of the analysis. In this work, we specifically focus on edge homophily, the simplest and most commonly used measure, which quantifies the fraction of edges that connect nodes with the same label.

**Homophily and GNN.** Recent studies have highlighted the performance degradation of GNNs on non-homophilic graphs (Zheng et al., 2022; Zhu et al., 2020; Yan et al., 2022; Zhu et al., 2021; Wang et al., 2022), often demonstrated through empirical evaluations. To address this challenge, various GNN architectures have been designed to better handle heterophilic graph structures (Zheng et al., 2022). For instance, MixHop (Abu-El-Haija et al., 2019) extends standard GNNs by aggregating information from multi-hop neighbors, which may help in dealing with heterophily. Similarly, GPRGNN (Chien et al., 2020) introduces an adaptive learning mechanism for Generalized PageRank (GPR) weights, enabling a more effective integration of node features and structural information. $H_2$GCN (Zhu et al., 2020) further refines message passing by leveraging higher-order connectivity patterns to enhance performance on heterophilic graphs. CAGNN (Chen et al., 2023) separates node features for prediction and aggregation, using a shared mixer to adaptively integrate neighbor information.

**Graph rewiring.** Graph rewiring improves GNN learning by modifying the edge set (Attali et al., 2024), addressing challenges like over-smoothing and over-squashing (Nguyen et al., 2023; Topping et al., 2021). Rewiring can also enhance edge homophily, but few works have explored this, with DHGR (Bi et al., 2024) being the most notable. Most rewiring algorithms rely on predefined structural criteria for edge addition or deletion, rather than learning a new graph structure. In contrast, the DHGR approach involves learning a similarity measure between nodes and solving an optimization problem. While presented as a rewiring method, it aligns more closely with Graph Structure Learning (GSL) (Zhou et al., 2023), which optimizes graph topology. However, our method offers a

simpler and more efficient approach to enhancing homophily, avoiding complex optimization while providing theoretical guarantees on the homophily of the rewired graph.

## 3 NOTATION

A graph is denoted by $\mathcal{G} = (\mathcal{V}, \mathcal{E})$, where $\mathcal{V}$ is the set of nodes and $\mathcal{E}$ is the set of edges. The node features of a graph are organized in a matrix $\mathbf{X} \in \mathbb{R}^{n \times d}$, where each row, denoted by $x_i$, is the $d$-dimensional feature vector of node $i$. We focus on node prediction tasks, thus node labels are available and denoted by $\mathbf{Y} \in \mathbb{R}^{n \times 1}$, where each element, denoted by $y_i$, is the label of node $i$. For simplicity, we assume scalar labels.

The edge homophily of a graph $\mathcal{G}$ is defined as:

$$H(\mathcal{G}) = \frac{|\{(u,v) \mid u,v \in \mathcal{V}, y_u = y_v\}|}{|\mathcal{E}|}.$$

Let $\mathcal{G}^c = (\mathcal{V}, \mathcal{E}^c)$ denote the complementary graph of $\mathcal{G}$, where the edge set $\mathcal{E}^c$ is given by $\mathcal{E}^c = \{(u,v) \mid u,v \in \mathcal{V}, (u,v) \notin \mathcal{E}\}$. Given two graphs $\mathcal{G}_1 = (\mathcal{V}, \mathcal{E}_1)$ and $\mathcal{G}_2 = (\mathcal{V}, \mathcal{E}_2)$, their union graph is denoted by $\mathcal{G}_1 \cup \mathcal{G}_2 = (\mathcal{V}, \mathcal{E}_1 \cup \mathcal{E}_2)$ and their intersection graph by $\mathcal{G}_1 \cap \mathcal{G}_2 = (\mathcal{V}, \mathcal{E}_1 \cap \mathcal{E}_2)$, and their residual graph by $\mathcal{G}_1 \setminus \mathcal{G}_2 = (\mathcal{V}, \mathcal{E}_1 \setminus \mathcal{E}_2)$, containing the edges in $\mathcal{G}_1$ that are not in $\mathcal{G}_2$.

## 4 CONTROLLING HOMOPHILY WITH A REFERENCE GRAPH

Empirical studies have shown that GNNs perform well on homophilic graphs, whereas heterophilic graphs pose significant challenges for standard GNN architectures (Zheng et al., 2022; Zhu et al., 2020; Yan et al., 2022; Zhu et al., 2021; Wang et al., 2022). Before presenting our framework for controlling graph homophily, we further support these empirical observations by making the relationship between homophily and node representation learning formal using the smoothness of the node embedding.

Using the standard measure of signal smoothness on a graph (Kalofolias, 2016), also known as the Dirichlet energy, the smoothness of the node embeddings is given by $\text{tr}(\mathbf{Z}^T \mathcal{L} \mathbf{Z})$, where $\mathbf{Z} \in \mathbb{R}^{n \times d}$ represents the learned node embeddings, $n$ is the number of nodes, $d$ is the embedding dimension, and $\mathcal{L} = \mathbf{D} - \mathbf{A}$ is the graph Laplacian, with $\mathbf{A}$ being the adjacency matrix and $\mathbf{D}$ the degree matrix. This measure captures how smoothly the embeddings vary across the graph structure, where small values indicate small changes between connected nodes (smoothness) and large values indicate large changes between connected nodes (lack of smoothness).

It is well established that the message passing mechanism in GNNs tends to produce smooth node embeddings ("smoothing is the nature of GNNs" (Chen et al., 2020)). That is, GNNs inherently reduce the smoothness term $\text{tr}(\mathbf{Z}^\top \mathcal{L} \mathbf{Z})$, which is not always beneficial and could lead to over-smoothing. In practice, the quality of node embeddings is often evaluated by their separability, as it reflects how well the nodes can be correctly classified, with linear separability serving as a practical criterion. The following result shows that the ability of GNNs to generate such linearly separable, and therefore effective, node embeddings improves as the homophily of the graph increases.

**Theorem 1.** *Let $\mathcal{G}$ be a graph with linearly separable node embeddings $\mathbf{Z}$. We have:*

$$tr(\mathbf{Z}^T \mathcal{L} \mathbf{Z}) \geq \frac{\alpha_m |\mathcal{E}|}{2\|\mathbf{W}\|^2} \left(1 - H(\mathcal{G})\right),$$

*where $\mathbf{W}$ are the parameters of a linear classifier separating $\mathbf{Z}$, $\alpha_m = \min_{(u,v) \in \mathcal{E}} A_{u,v}$, $\mathbf{A}$ is the adjacency matrix, and $|\mathcal{E}|$ is the number of edges in the graph.*

The proof of this result along with the proofs of all the other results in this section are in Appendix A.

The right-hand side of the inequality provides a lower bound on the smoothness of linearly separable embeddings, which is inversely related to the graph's homophily. Specifically, higher homophily results in a smaller lower bound, making it easier for GNNs to produce embeddings that are both smooth and linearly separable. Conversely, in heterophilic graphs, the increased lower bound raises

the risk that the smoothing induced by message passing may compromise linear separability. Thus, Thm. 1 highlights a key insight: *the greater the homophily of the graph, the higher the potential of a GNN to learn linearly separable, and thus more effective, node embeddings.*

### 4.1 HOMOPHILY-ENHANCING REWIRING FRAMEWORK

Building on Thm. 1, we propose a framework to enhance the homophily of a graph $\mathcal{G}$ using a reference graph, which we define as $\mathcal{G}_r = (\mathcal{V}, \mathcal{E}_r)$. This reference graph shares the same node set $\mathcal{V}$ as the original graph $\mathcal{G}$, but has a distinct edge set $\mathcal{E}_r$. Our approach leverages edge addition and deletion, standard practices in graph rewiring, with the key distinction that these operations are guided by the reference graph $\mathcal{G}_r$. Under specific conditions dependent on $\mathcal{G}_r$, we demonstrate that this rewiring process guarantees an improvement in the homophily of the resulting rewired graph $\mathcal{G}^{(k)}$, where $k \in \mathbb{Z}$ indicates the number of added or deleted edges. In Section 5, we detail how to construct a useful reference graph with the underutilized node features and labels.

Given a reference graph $\mathcal{G}_r = (\mathcal{V}, \mathcal{E}_r)$, the rewired graph $\mathcal{G}^{(k)} = (\mathcal{V}, \mathcal{E}^{(k)})$ is obtained by modifying the edge set $\mathcal{E}$ through the addition or deletion of $k$ edges based on $\mathcal{E}_r$. Specifically, $\mathcal{E}^{(k)}$ is defined as:

$$
\mathcal{E}^{(k)} = \begin{cases} \mathcal{E} \cup S_k, & \text{if } k > 0, \\ \mathcal{E} \setminus S_{|k|}, & \text{if } k < 0, \end{cases} \tag{1}
$$

where $S_k$ is a random subset of $k$ edges from $\mathcal{E}_r \setminus \mathcal{E}$ (edges in $\mathcal{G}_r$ but not in $\mathcal{G}$), and $S_{|k|}$ is a random subset of $|k|$ edges from $\mathcal{E} \cap \mathcal{E}_r^c$ (edges in both $\mathcal{G}$ and $\mathcal{G}_r^c$). Here, $k > 0$ indicates edge addition, and $k < 0$ edge deletion.

**Edge addition.** The following proposition and corollary describe the impact of adding $k$ edges selected at random from $\mathcal{E}_r \setminus \mathcal{E}$ on the homophily of the rewired graph depending on the homophily of the reference graph. For any $k > 0$, let $\mathcal{G}^{(k)}$ be the graph obtained by adding $k$ random edges from $\mathcal{E}_r \setminus \mathcal{E}$ to $\mathcal{G}$, and let $\mathcal{G}^{(k+1)}$ be the graph obtained by adding $k + 1$ such edges. The expected change in homophily stemming from this addition is described in the following result.

**Proposition 1.** *If $H(\mathcal{G}_r \setminus \mathcal{G}) > H(\mathcal{G})$, then*

$$
\mathbb{E}[H(\mathcal{G}^{(k+1)})] > \mathbb{E}[H(\mathcal{G}^{(k)})] > H(\mathcal{G}).
$$

*Otherwise,*

$$
\mathbb{E}[H(\mathcal{G}^{(k+1)})] \le \mathbb{E}[H(\mathcal{G}^{(k)})] \le H(\mathcal{G}).
$$

**Corollary 1.** *If $|\mathcal{E}_r| >> |\mathcal{E}|$, then $H(\mathcal{G}_r) \approx H(\mathcal{G}_r \setminus \mathcal{G})$, and the condition in Prop. 1 simplifies to $H(\mathcal{G}_r) > H(\mathcal{G})$.*

**Edge deletion.** The following proposition is similar to Prop. 1 but for edge deletion, i.e., where $k < 0$ and the rewired graph $\mathcal{G}^{(k)}$ is obtained by deleting $|k|$ edges randomly selected from $\mathcal{E} \cap \mathcal{E}_r^c$. The expected change in homophily stemming from this deletion is described in the following result.

**Proposition 2.** *If $H(\mathcal{G} \cap \mathcal{G}_r^c) < H(\mathcal{G})$, then*

$$
\mathbb{E}[H(\mathcal{G}^{(k-1)})] > \mathbb{E}[H(\mathcal{G}^{(k)})] > H(\mathcal{G}).
$$

*Otherwise,*

$$
\mathbb{E}[H(\mathcal{G}^{(k-1)})] \le \mathbb{E}[H(\mathcal{G}^{(k)})] \le H(\mathcal{G}).
$$

Thus, when their respective conditions hold, both edge addition and deletion guided by the reference graph improve homophily in expectation. See Appendix A.4 for concentration bounds and quantitative estimates of the improvement in homophily for both cases.

### 4.2 VALIDATION ON REAL-WORLD DATASETS

For validation, we consider several real-world datasets. For each dataset, we consider two graphs. The first, denoted by $\mathcal{G}$, is the original graph obtained from the dataset. The second graph is the "ideal" reference graph with maximum homophily, i.e., where nodes of the same class are connected

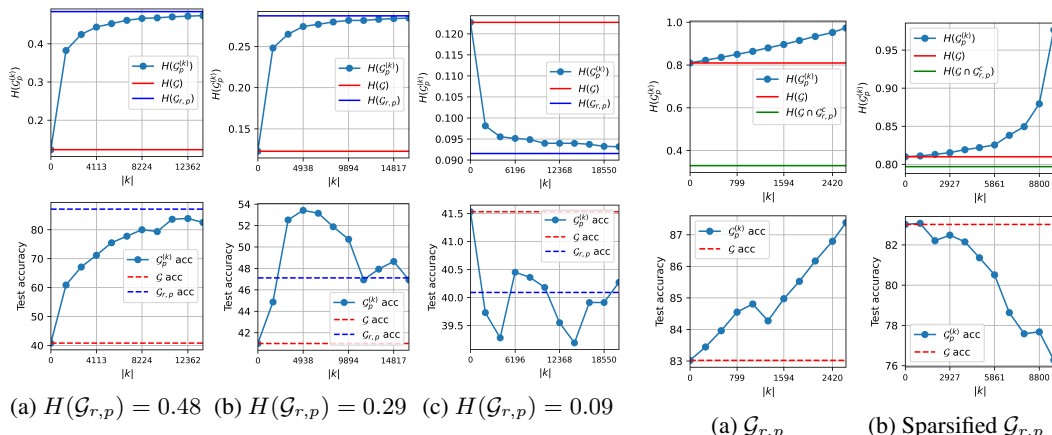

(a) $H(\mathcal{G}_{r,p}) = 0.48$ (b) $H(\mathcal{G}_{r,p}) = 0.29$ (c) $H(\mathcal{G}_{r,p}) = 0.09$

(a) $\mathcal{G}_{r,p}$ (b) Sparsified $\mathcal{G}_{r,p}$

Figure 1: Edge addition on Cornell: First row shows homophily increases with $k$ when the condition holds (blue above red); second row shows its impact on GCN accuracy. $H(\mathcal{G}_{r,p})$ decreases from left to right across the columns.

Figure 2: Edge deletion on Cora: Homophily increases when the condition is met (green line below red).

and nodes of different classes are disconnected. Since we do not have access to all the labels and cannot build such an ideal reference graph in practice, we generate a set of reference graphs, denoted by $\mathcal{G}_{r,p}$, using the following scheme. To control the homophily of the reference graphs, we use a parameter $p \in (0,1)$ representing the probability of randomly disconnecting or connecting edges of the ideal reference graph. As $p$ increases, the homophily of the modified reference graph $\mathcal{G}_{r,p}$ decreases. In addition, we denote by $\mathcal{G}_{r,p}^c$ the complement of $\mathcal{G}_{r,p}$. For each value of $p$, we rewire $\mathcal{G}$ by adding ($k > 0$) or deleting ($k < 0$) $|k|$ edges based on the reference graph $\mathcal{G}_{r,p}$ following Eq. 1, resulting in the rewired graph $\mathcal{G}_p^{(k)}$. When the condition of Prop. 1 is met, or equivalently when $|\mathcal{E}_r| \gg |\mathcal{E}|$ (Cor. 1), edge addition is expected to improve homophily. In practice, we use Cor. 1, as its assumption holds under the construction of $\mathcal{G}_{r,p}$. Similarly, when the condition of Prop. 2 is met, edge deletion is expected to improve homophily. To validate this, we measure and present the obtained empirical homophily of $\mathcal{G}_p^{(k)}$ and evaluate node classification accuracy using GCN, averaging results over 30 runs for each $p$ and $k$.

Figure 1 shows the effect of rewiring with edge addition on the Cornell dataset, where each column represents rewiring using a different $\mathcal{G}_{r,p}$ with a different homophily (controlled by different values of $p$). The first row displays homophily, $H(\mathcal{G}_p^{(k)})$, as a function of $k$, where the red and blue horizontal dashed lines indicate $H(\mathcal{G})$ and $H(\mathcal{G}_{r,p})$, respectively. The second row shows the impact on GCN node classification accuracy ($\mathcal{G}_p^{(k)}$ acc), where the red and blue dashed lines represent the accuracy obtained by $\mathcal{G}$ and $\mathcal{G}_{r,p}$, respectively. In Subfigures 1a and 1b, homophily increases with $k$. This aligns with Cor. 1, as the condition is met – the homophily of the reference graph indicated by the blue line is larger than the homophily of the original graph indicated by the red. Conversely, in Subfigure 1c, where the condition is not met (blue line below red), homophily decreases, further supporting the corollary. In the classification accuracy plots, where $H(\mathcal{G}_{r,p})$ is much higher than $H(\mathcal{G})$ (1a), performance improves with increasing $|k|$, but remains lower than training directly on $\mathcal{G}_{r,p}$. With $H(\mathcal{G}_{r,p})$ moderately higher than $H(\mathcal{G})$(1b), accuracy improves up to a point, after which over-smoothing degrades performance. Similar trends are observed in other datasets (see Appendix B). When the condition is not met (1c), edge addition reduces both homophily and performance.

Figure 2 shows the effect of edge deletion on the Cora dataset. The first row displays $H(\mathcal{G}_p^{(k)})$ as a function of $k$, where the red and green horizontal lines represent $H(\mathcal{G})$ and $H(\mathcal{G} \cap \mathcal{G}_{r,p})$, respectively. The second row shows the impact on GCN node classification accuracy. We observe that $H(\mathcal{G}_p^{(k)})$ aligns with Prop. 2: removing edges increases homophily when the condition is met (green line below red), and decreases it otherwise. However, higher homophily does not always improve GCN performance, as over-squashing can occur. In one scenario (2a), edge deletion based on $\mathcal{G}_{r,p}$ with $p = 0.1$ improves performance. In the second scenario (2b), we see that a sparse version of the same

$\mathcal{G}_{r,p}$ (with 90% edges removed) leads to performance degradation due to the excessive removal of same-class edges, raising the risk of over-squashing (despite the improved homophily).

# 5 PROPOSED REWIRING METHOD

Given a graph $\mathcal{G} = (\mathcal{V}, \mathcal{E})$, we assume, without loss of generality, that the nodes $\{1, \ldots, \overline{n}\}$, where $\overline{n} < n$, are labeled (training set), while the nodes $\{\overline{n} + 1, \ldots, n\}$ are unlabeled (validation and test sets). Let $\mathbf{X} \in \mathbb{R}^{n \times d}$ denote the node feature matrix, where $x_i \in \mathbb{R}^d$ represents the feature vector of node $i$, and let $\overline{\mathbf{Y}} = \in \mathbb{R}^{\overline{n} \times 1}$ denote the labels of the training nodes, where $\overline{y}_i$ is the scalar label of node $i$. The goal is to construct a homophilic reference graph $\mathcal{G}_r$, which, when applied in the rewiring framework outlined in Section 4, improves the homophily of the resulting rewired graph, $H(\mathcal{G}^{(k)})$, in comparison to the original graph, $H(\mathcal{G})$. If all labels were available, we could construct the ideal reference graph as in Subsection 4.2. But with access only to training labels $\overline{\mathbf{Y}}$, we aim to construct a reference graph whose homophily approaches maximal homophily. Naïvely building the reference graph solely based on $\overline{\mathbf{Y}}$ – by connecting each pair of training nodes that share the same class label – limits generalization and degrades performance when used for rewiring (see Section 6.1), necessitating a more robust approach. Assuming that the feature space offers a meaningful measure of similarity that aligns with the labels, a reference graph based on feature affinities should be homophilic. Our key idea is to combine node features with the training labels to construct a reference graph that is not only homophilic but also generalizes to the unlabeled nodes.

To implement this idea, we use the label-driven diffusion approach introduced in Mendelman & Talmon (2025) and propose a diffusion-based method to "complete" the missing labels by propagating label information to the unlabeled nodes through a graph constructed from the fully available node features $\mathbf{X}$. This method captures the shared structure between features and labels, resulting in a reference graph with higher homophily than the one constructed solely from $\mathbf{X}$ in most cases, as we empirically demonstrate in Appendix E.3. We claim that this diffusion-based construction results in a homophilic reference graph that is well-suited for our rewiring framework, and we support this claim with extensive empirical results presented in Section 6. While our framework supports any $\mathcal{G}_r$ that satisfies the conditions for improving homophily, the construction proposed here is one possible choice among others.

The first step in constructing the reference graph $\mathcal{G}_r$ is to build an affinity matrix $\mathbf{W}_D \in \mathbb{R}^{n \times n}$ based on the node feature vectors, where the elements are given by the Gaussian kernel $\mathbf{W}_D(i, j) = \exp\left(-\frac{d^2(x_i, x_j)}{\epsilon}\right)$, for $i, j \in \{1, \ldots, n\}$. Here, $d(\cdot, \cdot)$ is a distance metric in $\mathbb{R}^d$ (e.g., Euclidean distance), and $\epsilon$ is a hyperparameter that controls the scale of the affinity.

Next, we normalize the affinity matrix to obtain the data kernel $\mathbf{D}$, using a standard kernel normalization procedure (Mendelman & Talmon, 2025; Coifman & Lafon, 2006). We compute a diagonal matrix $\mathbf{D}_1$ whose diagonal elements consist of the sum of the rows of $\mathbf{W}_D$ and use it to obtain the intermediate matrix $\widetilde{\mathbf{D}} = \mathbf{D}_1^{-1} \mathbf{W}_D \mathbf{D}_1^{-1}$. Then, we compute another diagonal matrix $\mathbf{D}_2$ consisting of the row sums of $\widetilde{\mathbf{D}}$ and apply a second normalization step, yielding the data kernel $\mathbf{D} = \mathbf{D}_2^{-\frac{1}{2}} \widetilde{\mathbf{D}} \mathbf{D}_2^{-\frac{1}{2}}$.

For the label-based affinity matrix, since the labels of the validation and test sets are unknown, we define the following binary matrix $\mathbf{W}_P \in \mathbb{R}^{n \times n}$:

$$\mathbf{W}_P(i, j) = \begin{cases} 1, & \text{if } i, j \leq \overline{n} \text{ and } \overline{y}_i = \overline{y}_j, \text{ or } i = j, \\ 0, & \text{otherwise.} \end{cases} \tag{2}$$

Here for simplicity, we assume categorical labels for node classification, but $\mathbf{W}_P$ can be constructed based on continuous labels for graph regression with label affinities. This matrix $\mathbf{W}_P$ is then normalized using the same normalization applied to $\mathbf{W}_D$, yielding the label kernel $\mathbf{P}$.

We consider the following product of the kernels $\mathbf{\Gamma} = \mathbf{PDP}$, representing a label-driven diffusion process with three consecutive steps: propagation within classes using available labels, diffusion via node feature similarity across all nodes, and a final propagation through labels. This process uses node features to "complete" missing labels by propagating label information to unlabeled nodes, capturing the shared geometry between the node features and labels, as demonstrated in Mendelman & Talmon (2025). See Appendix C.1, where we visualize the difference between $\mathbf{PDP}$ and $\mathbf{D}$.

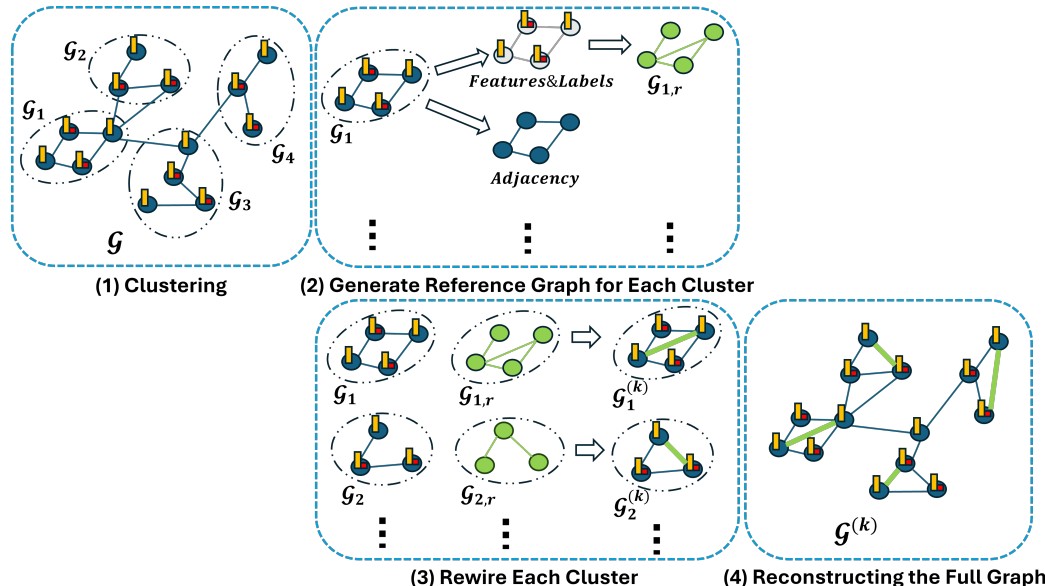

Figure 3: Rewiring method overview: (1) Cluster the original graph into clusters. (2) Construct a reference graph for each cluster using node features (yellow rectangles) and available labels (red squares). (3) Rewire each cluster by modifying edges based on its reference graph. (4) Reconstruct the fully rewired graph by incorporating inter-cluster edges. We denote the $i$-th cluster as $\mathcal{G}_i$, its reference graph as $\mathcal{G}_{i,r}$, and the rewired cluster as $\mathcal{G}_i^{(k)}$.

Finally, we use $\mathbf{\Gamma}$ to define the edge set of the reference graph $\mathcal{E}_r$. Since the elements of $\mathbf{\Gamma}$ are continuous, we clip them to obtain a binary matrix that corresponds to the adjacency matrix of an unweighted graph. To this end, we first compute the mean of each row in the matrix $\mathbf{\Gamma}$, given for the $i$-th row by $\mu_i = \frac{1}{n}\sum_{j=1}^n \mathbf{\Gamma}(i,j)$. We then clip the elements of each row $i$ based on the mean value $\mu_i$, resulting in the clipped kernel $\hat{\mathbf{\Gamma}}$:

$$\hat{\mathbf{\Gamma}}(i,j) = \begin{cases} 0, & \text{if } \mathbf{\Gamma}(i,j) < \mu_i, \\ 1, & \text{if } \mathbf{\Gamma}(i,j) \geq \mu_i, \end{cases} \quad (3)$$

The binary matrix $\hat{\mathbf{\Gamma}}$ defines the edge set of $\mathcal{G}_r$, given by $\mathcal{E}_r = \{(i,j) \mid \hat{\mathbf{\Gamma}}(i,j) > 0\}$.

---

**Algorithm 1** REFine

**Input:** graph $\mathcal{G}$, scale parameter $\epsilon$, cluster size $c$, # added/deleted edges per cluster $k$

Partition $\mathcal{G}$ to $N$ clusters
**for** $i = 1$ **to** $N$ **do**
  1: Construct $\mathbf{\Gamma} = \mathbf{PDP}$ from data and labels
  2: Clip $\mathbf{\Gamma}$ to obtain $\hat{\mathbf{\Gamma}}$ (Eq. 3)
  3: Obtain the edge set $\mathcal{E}_{l,r}$ from $\hat{\mathbf{\Gamma}}$
  4: Obtain $\mathcal{E}_l^{(k)}$ using $\mathcal{E}_l$ and $\mathcal{E}_{l,r}$ (Eq. 1)
**end for**
Reconstruct the full rewired graph $\mathcal{G}^{(k)}$

---

After constructing $\mathcal{G}_r$, it is used to rewire $\mathcal{G}$ by adding or deleting $k$ edges, as described in Section 4. Importantly, improvement in homophily is guaranteed only when the conditions in Prop. 1 and/or Prop. 2 are satisfied. To check if these conditions hold, we approximate $H(\mathcal{G})$ and $H(\mathcal{G}_r)$ using the available training and validation labels (validation labels are used solely for verifying homophily, not as part of the method). For a demonstration of the approximation's effectiveness, see Appendix C.3.

Given the obtained reference graph $\mathcal{G}_r$, we rewire the original graph $\mathcal{G}$ following the framework from Section 4. Edge addition ($k > 0$) is performed by randomly selecting $k$ edges from $\mathcal{E}_r \setminus \mathcal{E}$ and adding them to $\mathcal{G}$. Similarly, for edge deletion ($k < 0$), we remove $|k|$ randomly selected edges from $\mathcal{E} \cap \mathcal{E}_r^c$ using the complement graph $\mathcal{G}_r^c$. The parameter $k$, representing the number of edges added or deleted, is treated as a hyperparameter in our method. This results in the rewired graph $\mathcal{G}^{(k)}$.

**Scaling up.** As our method relies on kernel operations, it is costly on large graphs. To ensure scalability, we cluster the graph $\mathcal{G}$ into $N = \frac{|\mathcal{V}|}{c}$ balanced clusters using the METIS algorithm (Karypis & Kumar, 1998), where $c$ is the cluster size treated as a hyperparameter. This yields the set

Table 1: Results on node-classification datasets comparing None (no rewiring), SDRF, FoSR, BORF, and REFine. Accuracy is reported for all datasets; for Tolokers and Questions we report ROC AUC due to class imbalance, following Platonov et al. (2023). "T/O" = timeout; "OOM" = out of memory. Best **bold**; second-best underlined. For REFine, ↑/↓ show the sign of the gain vs. the best baseline. See Appendix E.1 for the complete table with SEM.

| Dataset | GCN | | | | | GATv2 | | | | | APPNP | | | | |
|---|---|---|---|---|---|---|---|---|---|---|---|---|---|---|---|
| Nodes, $H(\mathcal{G})$ | None | SDRF | FoSR | BORF | REFine | None | SDRF | FoSR | BORF | REFine | None | SDRF | FoSR | BORF | REFine |
| Cornell
183,0.12 | 51.8 | 58.4 | 51.6 | 53 | **71.3**
↑12.9 | 43.7 | 51 | 46 | 44.6 | **74**
↑23 | 49.4 | 63.7 | 55.1 | 55.1 | **74.6**
↑10.9 |
| Texas
183,0.06 | 59.7 | 65.4 | 62.4 | 62.1 | **79.1**
↑13.7 | 53.2 | 61.8 | 59.7 | 55.1 | **82.4**
↑20.6 | 61.9 | 77 | 67 | 65.1 | **82.4**
↑5.4 |
| Wisconsin
251,0.17 | 57.2 | 68.6 | 60.5 | 56.3 | **82.5**
↑13.9 | 53.3 | 63.3 | 60.9 | 52.5 | **84.9**
↑21.6 | 62.1 | 75 | 68.4 | 66 | **86**
↑11 |
| Chameleon
851,0.23 | 41.3 | 40.6 | 43.1 | 41.6 | **44.1**
↑1 | 40.8 | 39.5 | 40.1 | 41.2 | **43.5**
↑2.3 | 40.2 | 41 | 41.8 | 39.6 | **44.5**
↑2.7 |
| Squirrel
2223,0.2 | 40.7 | **41.5** | 39.7 | 40.3 | 41.1
↓0.4 | 37.4 | 37.7 | 37.7 | 36.7 | **38.8**
↑1.1 | 35.4 | 35.6 | 35.7 | 36.2 | **38.8**
↑2.6 |
| BlogCatalog
5196,0.4 | 77.6 | 77.9 | 77.4 | 78 | **85.2**
↑7.2 | 80.4 | 83.3 | 81.6 | 82.2 | **85.9**
↑2.6 | 95.7 | 95.8 | **95.9** | 95.5 | 95.7
↓0.2 |
| Actor
7160,0.21 | 28.4 | 29.2 | 28.1 | 28.3 | **31.3**
↑2.1 | 29.6 | 29.7 | 29.2 | 28.6 | **35.1**
↑5.4 | 33.8 | 33.8 | 33.9 | 33.6 | **34.8**
↑0.9 |
| BGP
10k,0.28 | 53.4 | 53.9 | 53.3 | 52 | **59.3**
↑5.4 | 62.3 | 63.2 | 62.8 | 63 | **63.3**
↑0.1 | 63.6 | 63.6 | 63.6 | 63.4 | **64.3**
↑0.7 |
| Tolokers
11k,0.59 | 77.2 | 77.6 | 77.4 | 77 | **78**
↑0.4 | 79.3 | **79.9** | 79.5 | 79.4 | 79.7
↓0.2 | 71.1 | 71.8 | 71.9 | 71.4 | **73.8**
↑1.9 |
| RomanEmpire
22k,0.04 | 37 | 46.2 | 36.9 | 35.2 | **58.8**
↑12.6 | 14.8 | 20.8 | 14.7 | 14.9 | **28.5**
↑7.7 | 14 | 22.6 | 14.3 | 15.5 | **30.8**
↑8.2 |
| Questions
48k,0.84 | 65.7 | OOM | 63.3 | 65.9 | **70.3**
↑4.4 | 67.4 | OOM | **67.6** | **67.6** | 66.6
↓1 | 44.1 | OOM | 44.8 | 44.5 | **47**
↑2.2 |
| Elliptic
203k,0.71 | 87.1 | 87 | 85.9 | T/O | **89.5**
↑2.4 | 89.6 | 90.6 | 89.9 | T/O | **90.8**
↑0.2 | 87.4 | 87.4 | 86.7 | T/O | **89.8**
↑2.4 |
| Genius
421k,0.59 | 83.1 | OOM | 82.2 | T/O | **83.8**
↑0.7 | 81.7 | OOM | 81.2 | T/O | **83.6**
↑1.9 | 81.9 | OOM | 81.2 | T/O | **83.6**
↑1.7 |

Table 2: Test accuracy on heterophilic graphs for specialized GNNs vs. ST+REFine. Best is **bold**, second-best is underlined. See Appendix E.1 for the complete table with SEM.

| | Cornell | Texas | Wisconsin | Chameleon | Squirrel | BlogCatalog | Actor | BGP | Roman-empire |
|---|---|---|---|---|---|---|---|---|---|
| MixHop | 71.9 | 79.1 | 83.1 | 43.2 | 39.8 | OOM | **36.2** | 64.3 | 32.1 |
| H$_2$GCN | 73.2 | **82.7** | 82.3 | 41.8 | 40.4 | **96.4** | 30.3 | 64.9 | 34.3 |
| GPRGNN | 70.8 | 81 | 82.5 | 40.9 | 38.5 | 95.7 | 35.4 | **65** | 20.5 |
| OrderedGNN | 70.8 | 77.8 | 82.1 | 38 | 34.3 | 95.7 | 35.8 | **65** | 45.5 |
| ST+REFine (ours) | **74.6** | 82.4 | **86** | **44.5** | **41.1** | 95.7 | 35.1 | 64.3 | **58.8** |

of graphs $\{\mathcal{G}_l = (\mathcal{V}_l, \mathcal{E}_l)\}_{l=1}^{N}$. Each cluster $\mathcal{G}_l$ is then rewired independently based on its reference graph $\mathcal{G}_{l,r} = (\mathcal{V}_l, \mathcal{E}_{l,r})$, constructed using the procedure described above. Finally, the full graph is reconstructed by merging all rewired clusters while preserving the original inter-cluster edges.

We term our rewiring algorithm, which uses a reference graph for refining homophily, *REFine*. Its key steps are summarized in Algorithm 1 and illustrated in Figure 3.

# 6 EXPERIMENTS

Table 1 compares the node classification performance of our REFine[1] with several well-established rewiring methods: SDRF (Topping et al., 2021), FoSR (Karhadkar et al., 2022), and BORF (Nguyen et al., 2023), across multiple GNN architectures: GCN (Kipf & Welling, 2016), GATv2 (Brody et al., 2021), and APPNP (Gasteiger et al., 2018). We evaluate 13 datasets, ranging from small datasets with hundreds of nodes to large datasets with up to $421k$ nodes, each with varying levels of homophily. The table depicts both the number of nodes and the homophily for each dataset. Our REFine outperforms the baseline methods in most cases, significantly improving performance compared to the leading baseline. For the complete table, including the standard error of the mean (SEM), see Appendix E.1. Due to the high computational cost of the competing rewiring methods for large datasets, we adapted all compared methods to use the same clustering strategy as in our approach (Section 5). See Appendix E.1 for results on high-homophily datasets (Cora, Citeseer,

---

[1]Our code is available in the supplementary materials and will be released on GitHub upon publication.

Pubmed), where our method and the baselines showed no significant improvement over training on the original graph.

Since REFine enhances the homophily of a graph, it makes the graph more compatible with standard message passing GNNs. To demonstrate the practical benefit, we compare the performance of standard GNNs combined with REFine to that of specialized GNNs designed for heterophilic graphs. Table 2 presents a comparison of node classification performance on heterophilic graphs ($H(\mathcal{G}) < 0.5$). It compares the performance of the top-performing standard GNNs (GCN, GATv2, and APPNP) combined with REFine rewiring (denoted as ST+REFine) for each dataset, against well-established specialized GNNs for heterophilic graphs: MixHop (Abu-El-Haija et al., 2019), GPRGNN (Chien et al., 2020), H$_2$GCN (Zhu et al., 2020), and OrderedGNN (Song et al., 2023). Notably, on most datasets, standard GNNs with REFine either match or outperform the specialized models. For the complete table, including the standard error of the mean (SEM), see Appendix E.1.

In Appendix E.2, we compare the homophily of the original graph $\mathcal{G}$, the reference graph $\mathcal{G}_r$, and the rewired graph $\mathcal{G}^{(k)}$ across multiple datasets, demonstrating the effectiveness of our rewiring method in enhancing homophily. See Appendix D for additional implementation details, including parameter choices for our method and the baselines.

### 6.1 ADDITIONAL EXPERIMENTS

**Homophily and rewiring effectiveness.** In Appendix G.1, we empirically demonstrate that datasets with lower original homophily tend to show greater test accuracy gains from our rewiring method. This is likely because the reference graph typically has much higher homophily than the original in such cases, resulting in a significantly more homophilic rewired graph and thus better performance.

**Labels-only ablation.** When the reference graph is built solely from training labels ($\mathbf{\Gamma} = \mathbf{P}$), $H(\mathcal{G}_r) = 1$, so rewiring necessarily increases homophily. However, this affects only the training subgraph and fails to generalize, leading to worse performance. Appendix G.2 shows that with $\mathbf{\Gamma} = \mathbf{P}$, as $|k|$ increases, homophily improves as expected but test accuracy declines.

**Cluster size.** Larger clusters can yield small performance gains, but with increased runtime. Our experiments use cluster sizes of 100 or 500 depending on graph size. See Appendix G.3 for details.

## 7 COMPLEXITY AND RUNTIME

REFine has per-cluster complexity $\mathcal{O}(c^3)$, and with $n/c$ clusters this yields $\mathcal{O}(c^2 n)$, where $n$ is the number of nodes and $c$ is the cluster size. Including clustering with METIS (average-case $\mathcal{O}(|\mathcal{E}|)$), the end-to-end complexity is $\mathcal{O}(|\mathcal{E}| + c^2 n)$. On standard sparse benchmarks ($|\mathcal{E}| = \mathcal{O}(n)$), the complexity is $\mathcal{O}(c^2 n)$, and for large graphs with $n \gg c$ the method is effectively linear in $n$. The algorithm parallelizes across clusters, and with $g$ GPUs the parallel implementation runs in $\mathcal{O}((c^2/g)\,n)$. Appendix F presents complexity and runtime comparisons, demonstrating REFine's significant efficiency gains over existing methods.

## 8 CONCLUSION

In this paper, we introduced a rewiring framework that enhances graph homophily using a reference graph, with theoretical guarantees for increasing the homophily of the rewired graph. We validated the effectiveness of our framework for node classification on real-world datasets using synthetic reference graphs with controlled homophily. To extend its applicability, we proposed a label-driven diffusion method for constructing homophilic reference graphs from node features and training labels. While our method achieves strong empirical results, it is limited, like other homophily-enhancing approaches, by its inapplicability to graph classification and its reliance on the assumption that feature space offers a meaningful measure of similarity that aligns with the labels. Despite these limitations, our framework presents a systematic approach to enhancing graph homophily, leading to improved performance in node classification.

Future directions include exploring simultaneous edge addition and deletion, developing alternative strategies for constructing reference graphs, and leveraging reference graphs to enhance additional graph properties beyond homophily.

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

# A  PROOFS

## A.1  PROOF OF THEOREM 1

*Proof of Theorem 1.* Some of the steps of this proof follow Theorem 3.1 from (Xing et al., 2024), specifically adopting their definition of linearly separable embeddings.

To prove Theorem 1, we assume the existence of a linear classifier, parameterized by $\mathbf{W} \in \mathbb{R}^{d \times c}$, where $d$ is the embedding dimension and $c$ is the number of classes, which satisfies the condition $\mathbf{Y} = \mathbf{ZW}$.

Now, we express the term $\sum_{(u,v)\in\mathcal{E}} A_{u,v}\|\mathbf{y}_u - \mathbf{y}_v\|^2$ using the smoothness term:

$$
\sum_{(u,v)\in\mathcal{E}} A_{u,v}\|\mathbf{y}_u - \mathbf{y}_v\|^2 = \sum_{(u,v)\in\mathcal{E}} A_{u,v}\|\mathbf{z}_u\mathbf{W} - \mathbf{z}_v\mathbf{W}\|^2 \qquad \dots \mathbf{Y} = \mathbf{ZW}
$$

$$
= 2\|\mathbf{W}\|^2 \frac{1}{2} \sum_{(u,v)\in\mathcal{E}} A_{u,v}\|\mathbf{z}_u - \mathbf{z}_v\|^2
$$

$$
= 2\|\mathbf{W}\|^2 \mathrm{tr}(Z^T\mathcal{L}Z) \qquad \dots \text{Smoothness definition}
$$

where $\mathbf{y}_u$ denotes the one-hot label of node $u$.

Thus we get:

$$
\mathrm{tr}(Z^T\mathcal{L}Z) = \frac{1}{2\|\mathbf{W}\|^2} \sum_{(u,v)\in\mathcal{E}} A_{u,v}\|\mathbf{y}_u - \mathbf{y}_v\|^2
$$

Next, defining $\alpha_m = \min_{(u,v)\in\mathcal{E}} A_{u,v}$ as the minimum nonzero entry of $\mathbf{A}$, we obtain:

$$
\mathrm{tr}(Z^T\mathcal{L}Z) = \frac{1}{2\|\mathbf{W}\|^2} \sum_{(u,v)\in\mathcal{E}} A_{u,v}\|\mathbf{y}_u - \mathbf{y}_v\|^2
$$

$$
\geq \frac{\alpha_m}{2\|\mathbf{W}\|^2} \sum_{(u,v)\in\mathcal{E}} \|\mathbf{y}_u - \mathbf{y}_v\|^2.
$$

We now replace $\|\mathbf{y}_u - \mathbf{y}_v\|^2$ with the indicator function $I(\mathbf{y}_u \neq \mathbf{y}_v)$:

$$
\mathrm{tr}(Z^T\mathcal{L}Z) \geq \frac{\alpha_m}{2\|\mathbf{W}\|^2} \sum_{(u,v)\in\mathcal{E}} I(\mathbf{y}_u \neq \mathbf{y}_v)
$$

$$
= \frac{\alpha_m|\mathcal{E}|}{2\|\mathbf{W}\|^2} \left( \frac{1}{|\mathcal{E}|} \sum_{(u,v)\in\mathcal{E}} I(\mathbf{y}_u \neq \mathbf{y}_v) \right)
$$

$$
= \frac{\alpha_m|\mathcal{E}|}{2\|\mathbf{W}\|^2} \left( 1 - \frac{1}{|\mathcal{E}|} \sum_{(u,v)\in\mathcal{E}} I(\mathbf{y}_u = \mathbf{y}_v) \right)
$$

$$
= \frac{\alpha_m|\mathcal{E}|}{2\|\mathbf{W}\|^2} \left( 1 - H(\mathcal{G}) \right).
$$

$\square$

## A.2  PROOF OF PROPOSITION 1

*Proof of Proposition 1.* Let the graph $\mathcal{G}$ have $n + m$ edges, where $n$ edges connect nodes of the same label and $m$ edges connect nodes of different labels.

Let the residual graph $\mathcal{G}_r \setminus \mathcal{G}$ have $n' + m'$ edges, where $n'$ edges connect nodes of the same label and $m'$ edges connect nodes of different labels.

Define $x$ as the number of added edges between nodes of the same label:

$$x = \sum_{i=1}^{k} X_i,$$

where $X_i \sim \text{Bernoulli}\left(\frac{n'}{n'+m'}\right)$ are independent and identically distributed (i.i.d.).

The homophily rate of the rewired graph $\mathcal{G}^{(k)}$ is given by:

$$H(\mathcal{G}^{(k)}) = \frac{n+x}{n+m+k}.$$

Taking the expectation over the randomness of $\{X_i\}_{i=1}^{k}$, we have:

$$\mathbb{E}[H(\mathcal{G}^{(k)})] = \frac{n+\mathbb{E}[x]}{n+m+k}. \tag{4}$$

Now calculate $\mathbb{E}[x]$:

$$\mathbb{E}[x] = \mathbb{E}\left[\sum_{i=1}^{k} X_i\right] = \sum_{i=1}^{k} \mathbb{E}[X_i] = k \cdot \frac{n'}{n'+m'}.$$

Substitute $\mathbb{E}[x] = k \cdot \frac{n'}{n'+m'}$ into equation equation 4:

$$\mathbb{E}[H(\mathcal{G}^{(k)})] = \frac{n+k \cdot \frac{n'}{n'+m'}}{n+m+k}.$$

Taking the derivative of this expression with respect to $k$ yields:

$$\frac{\partial \mathbb{E}[H(\mathcal{G}^{(k)})]}{\partial k} = \frac{\frac{n'}{n'+m'} \cdot n + \frac{n'}{n'+m'} \cdot m - n}{(n+m+k)^2}.$$

The derivative is positive when $\frac{n'}{n'+m'} \cdot n + \frac{n'}{n'+m'} \cdot m - n > 0$. Reorganizing this inequality, we find:

$$\frac{n'}{n'+m'} > \frac{n}{n+m}.$$

Thus, when $H(\mathcal{G}_r \setminus \mathcal{G}) > H(\mathcal{G})$, the derivative of $\mathbb{E}[H(\mathcal{G}^{(k)})]$ with respect to $k$ is strictly positive, meaning that increasing the number of added edges increases $\mathbb{E}[H(\mathcal{G}^{(k)})]$. Thus,

$$\mathbb{E}[H(\mathcal{G}^{(k+1)})] > \mathbb{E}[H(\mathcal{G}^{(k)})] > \mathbb{E}[H(\mathcal{G}^{(0)})] = H(\mathcal{G}).$$

Conversely, the derivative is negative when $\frac{n'}{n'+m'} \cdot n + \frac{n'}{n'+m'} \cdot m - n < 0$, which implies:

$$\frac{n'}{n'+m'} < \frac{n}{n+m}.$$

or equivalently $H(\mathcal{G}_r \setminus \mathcal{G}) < H(\mathcal{G})$. In this case, the derivative is strictly negative, so the highest value of $\mathbb{E}[H(\mathcal{G}^{(k)})]$ occurs when $k = 0$. Thus,

$$\mathbb{E}[H(\mathcal{G}^{(k+1)})] < \mathbb{E}[H(\mathcal{G}^{(k)})] < \mathbb{E}[H(\mathcal{G}^{(0)})] = H(\mathcal{G}).$$

$\square$

## A.3 PROOF OF PROPOSITION 2

*Proof of Proposition 2.* For simplicity, we define $k > 0$, where $k$ represents the number of deleted edges.

Let the intersection graph $\mathcal{G} \cap \mathcal{G}_r^c$ have $n^* + m^*$ edges, where $n^*$ edges connect nodes of the same label and $m^*$ edges connect nodes of different labels.

Applying the same procedure as in the proof of Proposition 1, we get:

$$\mathbb{E}[H(\mathcal{G}^{(k)})] = \frac{n - k \cdot \frac{n^*}{n^* + m^*}}{n + m - k}.$$

Taking the derivative of this expression with respect to $k$ yields:

$$\frac{\partial \mathbb{E}[H(\mathcal{G}^{(k)})]}{\partial k} = \frac{-\frac{n^*}{n^* + m^*}(n + m) + n}{(n + m - k)^2}.$$

The derivative is positive when $-\frac{n^*}{n^* + m^*}(n + m) + n > 0$. Reorganizing this inequality, we find:

$$\frac{n^*}{n^* + m^*} < \frac{n}{n + m}.$$

This is equivalent to $H(\mathcal{G} \cap \mathcal{G}_r^c) < H(\mathcal{G})$. In this case, the derivative is strictly positive, so increasing the number of deleted edges increases $\mathbb{E}[H(\mathcal{G}^{(k)})]$. Thus, returning to the original notation where $k < 0$ represents deleting $|k|$ edges, we have:

$$\mathbb{E}[H(\mathcal{G}^{(k-1)})] > \mathbb{E}[H(\mathcal{G}^{(k)})] > \mathbb{E}[H(\mathcal{G}^{(0)})] = H(\mathcal{G}).$$

Conversely, the derivative is negative when $-\frac{n^*}{n^* + m^*}(n + m) + n < 0$, which implies:

$$\frac{n^*}{n^* + m^*} > \frac{n}{n + m}.$$

or equivalently $H(\mathcal{G} \cap \mathcal{G}_r^c) > H(\mathcal{G})$. In this case, the derivative is strictly negative, so the highest value of $\mathbb{E}[H(\mathcal{G}^{(k)})]$ occurs when $k = 0$, meaning no edges are deleted. Thus, returning to the original notation where $k < 0$ represents deleting $|k|$ edges, we have:

$$\mathbb{E}[H(\mathcal{G}^{(k-1)})] < \mathbb{E}[H(\mathcal{G}^{(k)})] < \mathbb{E}[H(\mathcal{G}^{(0)})] = H(\mathcal{G}).$$

$\square$

## A.4 CONCENTRATION BOUNDS AND MAGNITUDE OF IMPROVEMENT

The concentration bounds follow directly from the proofs of Propositions 1 and 2 via Hoeffding's inequality.

**Edge addition.** Let $x = \sum_{i=1}^{k} X_i$, where $X_i \sim$ Bernoulli $\left( \frac{n'}{n'+m'} \right)$ are i.i.d. random variables indicating whether an added edge is homophilic. Then, the homophily of the rewired graph is:

$$H(\mathcal{G}^{(k)}) = \frac{n+x}{n+m+k}$$

Using Hoeffding's inequality:

$$\mathbb{P}\left( |x - \mathbb{E}[x]| \geq \epsilon \right) \leq 2\exp\left( -\frac{2\epsilon^2}{k} \right)$$

This gives:

$$\mathbb{P}\left( \left| H(\mathcal{G}^{(k)}) - \mathbb{E}[H(\mathcal{G}^{(k)})] \right| \geq \delta \right) \leq 2\exp\left( -\frac{2\delta^2(n+m+k)^2}{k} \right)$$

and $|\mathcal{E}| = n + m$ so we get:

$$\mathbb{P}\left( \left| H(\mathcal{G}^{(k)}) - \mathbb{E}[H(\mathcal{G}^{(k)})] \right| > \delta \right) \leq 2\exp\left( -\frac{2\delta^2(|\mathcal{E}|+k)^2}{k} \right)$$

**Edge deletion.** Let $x = \sum_{i=1}^{k} X_i$, where $X_i \sim$ Bernoulli $\left( \frac{n^*}{n^*+m^*} \right)$ are i.i.d. random variables indicating whether a deleted edge is homophilic. Then, the homophily of the rewired graph is:

$$H(\mathcal{G}^{(k)}) = \frac{n-x}{n+m-k}$$

Using Hoeffding's inequality:

$$\mathbb{P}\left( |x - \mathbb{E}[x]| \geq \epsilon \right) \leq 2\exp\left( -\frac{2\epsilon^2}{k} \right)$$

This gives:

$$\mathbb{P}\left( \left| H(\mathcal{G}^{(k)}) - \mathbb{E}[H(\mathcal{G}^{(k)})] \right| \geq \delta \right) \leq 2\exp\left( -\frac{2\delta^2(n+m-k)^2}{k} \right)$$

and $|\mathcal{E}| = n + m$ so we get (defined only for $k < |\mathcal{E}|$):

$$\mathbb{P}\left( \left| H(\mathcal{G}^{(k)}) - \mathbb{E}[H(\mathcal{G}^{(k)})] \right| > \delta \right) \leq 2\exp\left( -\frac{2\delta^2(|\mathcal{E}|-k)^2}{k} \right)$$

These bounds confirm that for edge addition, when $|\mathcal{E}|^2 \gg k$, the homophily is tightly concentrated around its expectation and the concentration improves as $k$ increases. For edge deletion, when $|\mathcal{E}| \gg k$, the homophily is also tightly concentrated around its expectation, but the concentration becomes looser as $k$ increases.

The magnitude of improvement can also be derived directly from our proofs.

**Edge Addition.** From Proposition 1, the expected homophily after adding $k$ edges is:

$$\mathbb{E}[H(\mathcal{G}^{(k)})] = \frac{n + k \cdot \frac{n'}{n'+m'}}{n+m+k}, \quad H(\mathcal{G}) = \frac{n}{n+m}$$

where $n$ and $m$ denote the number of edges in $\mathcal{G}$ connecting nodes of the same and different classes, respectively, and $n'$ and $m'$ denote the corresponding counts in the reference graph $\mathcal{G}_r$.

Bringing to common denominator and simplifying:

$$\mathbb{E}[H(\mathcal{G}^{(k)})] - H(\mathcal{G}) = \frac{n - H(\mathcal{G})(|\mathcal{E}|+k) + kH(\mathcal{G}_r)}{|\mathcal{E}|+k}$$

by using $n = |\mathcal{E}|H(\mathcal{G})$ we get:

$$\mathbb{E}[H(\mathcal{G}^{(k)})] - H(\mathcal{G}) = \frac{|\mathcal{E}|H(\mathcal{G}) - H(\mathcal{G})(|\mathcal{E}|+k) + kH(\mathcal{G}_r)}{|\mathcal{E}|+k} = \frac{k(H(\mathcal{G}_r) - H(\mathcal{G}))}{|\mathcal{E}|+k}$$

**Edge Deletion.** From Proposition 2, the expected homophily after deleting $k$ edges is:

$$\mathbb{E}[H(\mathcal{G}^{(k)})] = \frac{n - k \cdot \frac{n^*}{n^* + m^*}}{n + m - k}, \quad H(\mathcal{G}) = \frac{n}{n + m}$$

where $n^*$ and $m^*$ denote the number of edges in $\mathcal{G} \cap \mathcal{G}_r^c$ connecting nodes of the same and different classes, respectively.

Bringing to common denominator and simplifying:

$$\mathbb{E}[H(\mathcal{G}^{(k)})] - H(\mathcal{G}) = \frac{n - H(\mathcal{G})(|\mathcal{E}| - k) - kH(\mathcal{G} \cap \mathcal{G}_r^c)}{|\mathcal{E}| - k}$$

by using $n = |\mathcal{E}|H(\mathcal{G})$ we get:

$$\mathbb{E}[H(\mathcal{G}^{(k)})] - H(\mathcal{G}) = \frac{k(H(\mathcal{G}) - H(\mathcal{G} \cap \mathcal{G}_r^c))}{|\mathcal{E}| - k}$$

# B ADDITIONAL SIMULATIONS

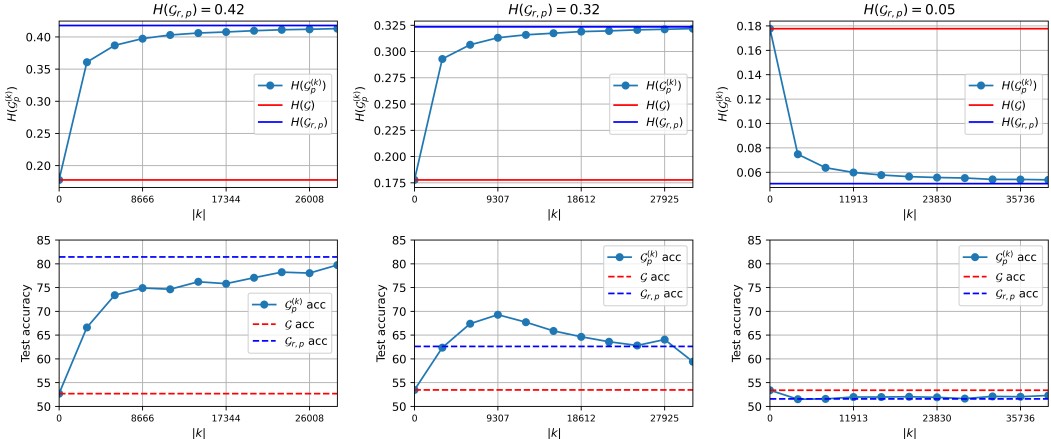

Figure 4: Simulation of edge addition on the Wisconsin dataset.

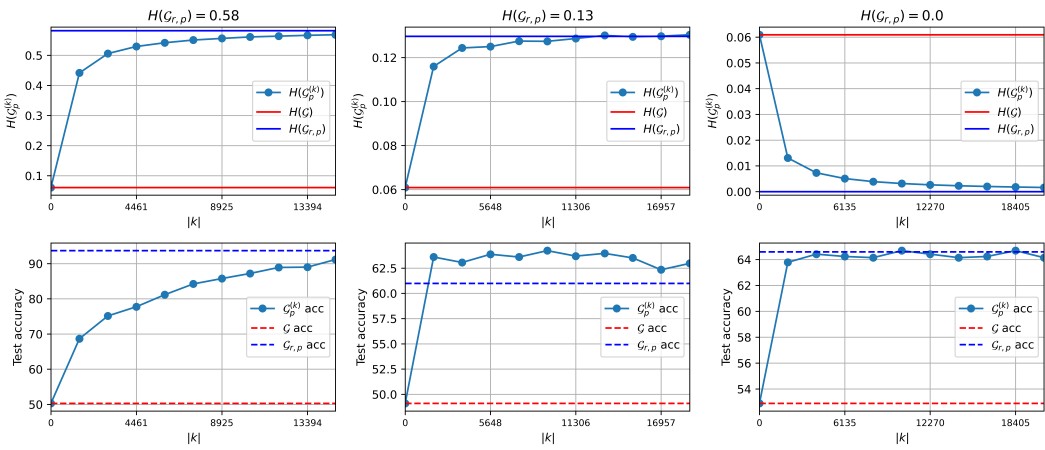

Figure 5: Simulation of edge addition on the Texas dataset.

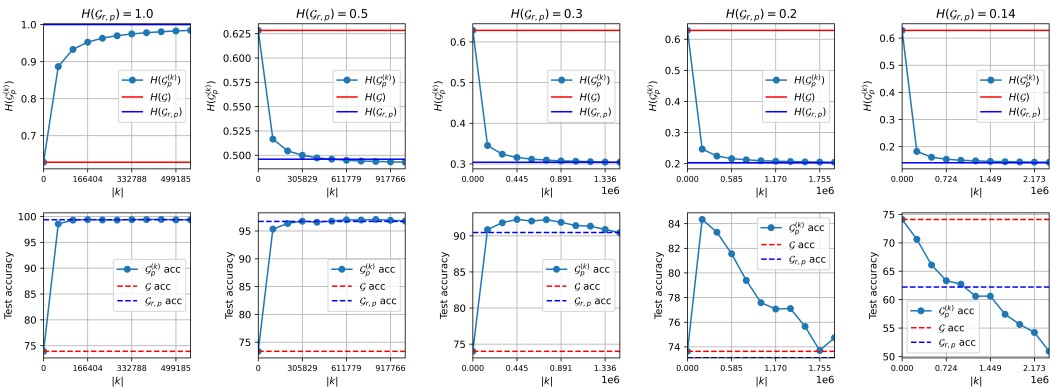

Figure 6: Simulation of edge addition on the Wiki dataset.

## C ADDITIONAL METHOD DETAILS

### C.1 VISUALIZATION OF KERNEL DIFFERENCES

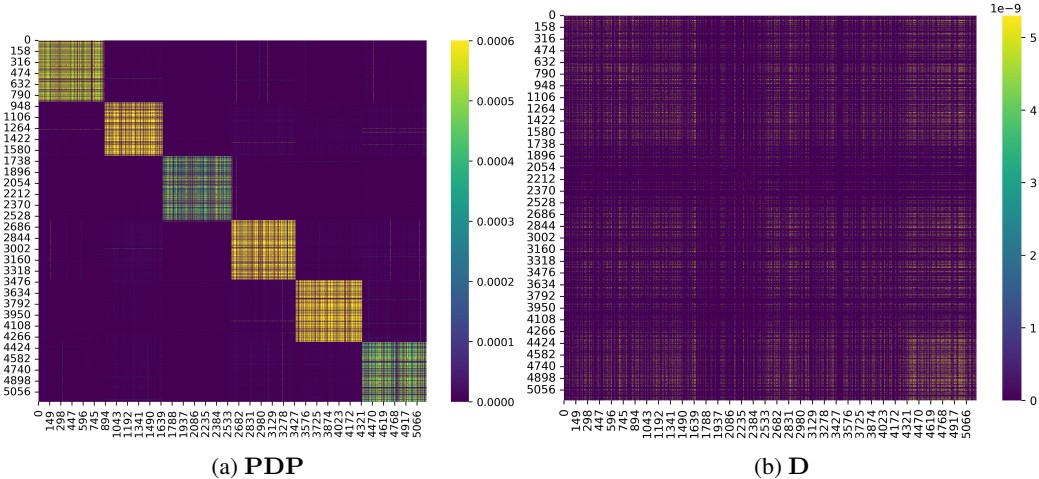

(a) **PDP**                                      (b) **D**

Figure 7: Heatmap visualizations of the kernels **PDP** and **D** for the BlogCatalog dataset. The rows and columns are sorted by label, with an ideal heatmap showing high-value diagonal blocks for each class. The **PDP** heatmap exhibits better class separation compared to **D**, reflecting improved structure.

### C.2 DIFFERENCES BETWEEN OUR GRAPH CONSTRUCTION AND LABEL-DRIVEN DIFFUSION

Our graph construction method using feature vectors and training labels differs from label-driven diffusion (Mendelman & Talmon, 2025) in two key ways: first, we apply a three-step diffusion process to ensure symmetry; second, we set inter-class distances to infinity in the label affinity kernel to promote intra-class connections.

### C.3 APPROXIMATING HOMOPHILY USING A SAMPLED GRAPH

To check if the conditions for enhancing homophily hold, we can approximate $H(\mathcal{G})$ and $H(\mathcal{G}_r)$ using the available training and validation labels (the validation labels are used solely for verifying homophily, not as part of the method). We construct a sampled graph $\mathcal{G}^s$ and sampled reference graph $\mathcal{G}_r^s$, using validation set nodes whose labels are withheld when constructing **P** but are known for evaluation. We randomly sample training nodes such that the ratio of unlabeled validation nodes to sampled labeled nodes matches the ratio of unlabeled (validation and test) nodes to labeled nodes in the full graph. The edge set of $\mathcal{G}^s$ consists of edges from $\mathcal{G}$ connecting pairs of nodes present in $\mathcal{G}^s$, and the edge set of $\mathcal{G}_r^s$ consists of edges from $\mathcal{G}_r$ connecting pairs of nodes present in $\mathcal{G}_r^s$. Since the probability of an edge connecting nodes of the same label should remain consistent between the graph and its sampled version, $H(\mathcal{G}^s)$ approximates $H(\mathcal{G})$ and $H(\mathcal{G}_r^s)$ approximates $H(\mathcal{G}_r)$. Thus, $\mathcal{G}^s$ and $\mathcal{G}_r^s$ allow us to check whether the conditions of Proposition 1 and/or Proposition 2 hold for $\mathcal{G}_r$, helping determine whether edge addition or deletion will improve the homophily of $\mathcal{G}^{(k)}$.

In Table 3, we report the approximated values on real-world datasets and demonstrate that the approximation closely reflects the true homophily. This validates the suitability of the sampled graphs for analyzing and verifying the assumptions underlying our rewiring framework.

Table 3: Comparison of true and sampled graph homophily values. The sampled graphs $\mathcal{G}^s$ and $\mathcal{G}_r^s$ provide a close approximation to the original homophily values $H(\mathcal{G})$ and $H(\mathcal{G}_r)$, respectively.

|  | Cornell | Texas | Wisconsin | BlogCatalog |
|---|---|---|---|---|
| $H(\mathcal{G})$ | 0.12 | 0.06 | 0.17 | 0.4 |
| $H(\mathcal{G}^s)$ | 0.14 | 0.08 | 0.2 | 0.4 |
| $H(\mathcal{G}_r)$ | 0.4 | 0.49 | 0.51 | 0.23 |
| $H(\mathcal{G}_r^s)$ | 0.41 | 0.48 | 0.5 | 0.23 |

## D  EXPERIMENT SETTINGS

All evaluated datasets are available in PyTorch-Geometric. When official data splits with at least five splits were provided, we used the official splits from PyTorch-Geometric. For datasets without official splits or with fewer than five, we randomly partitioned them into five train-validation-test splits (60%-20%-20%). For Chameleon and Squirrel, we used the filtered versions and corresponding splits provided by Platonov et al. (2023), as the original versions suffer from train-test data leakage.

For our REFine and all rewiring baselines, we apply our clustering strategy where, for graphs with fewer than 1,000 nodes, we set $c = |\mathcal{V}|$ (no clustering), for graphs with 1,000 to 25,000 nodes, we set $c = 500$, and for graphs with more than 25,000 nodes, we set $c = 100$.

We perform a grid search to optimize hyperparameters on the validation set and report test accuracy with standard error of the mean (SEM). The hidden dimension is set to 32, the learning rate is selected from $\{10^{-4}, 10^{-3}, 10^{-2}, 10^{-1}\}$, and weight decay is searched within $\{10^{-4}, 10^{-3}, 10^{-2}\}$. All models are trained using the Adam optimizer.

For all architectures, we used ReLU activation. Specifically, GCN consists of 2 GCN layers, GATv2 comprises 2 GATv2 layers, and APPNP includes 2 linear layers followed by an APPNP propagation layer with $K = 10$ and $\alpha = 0.1$.

MixHop consists of two MixHopConv layers, each using the default powers $[0, 1, 2]$, followed by a linear output layer. To accommodate the concatenation of three feature sets per layer, the hidden dimension is divided by 3. $H_2$GCN includes a linear feature embedding layer followed by two $H_2$GCNConv layers. Since each $H_2$GCNConv layer concatenates two embeddings, the hidden dimension is divided by 2. The final output layer is a linear projection applied to the concatenated features across all layers. GPRGNN consists of a 2-layer MLP followed by a GPR propagation module with the default $K = 10$, $\alpha = 0.1$, and initialization set to Random. OrderedGNN consists of an input linear transformation layer followed by two OrderedConv layers, each using a temporal matching module composed of a linear layer and layer normalization. The hidden dimension is divided into four chunks to enable chunk-wise updates. A final linear output layer is applied after the OrderedConv blocks.

We transform all directed datasets into undirected ones before applying rewiring and training, as METIS operates only on undirected graphs.

All experiments were conducted using Python on NVIDIA DGX A100 systems, each equipped with A100 GPUs and 512 GB of RAM, with T/0 reached after 24 hours of execution.

**REFine hyperparameters.** For the scale parameter $\epsilon$, we search for the optimal value in $\{1e-8, 1e-7, 1e-6, 1e-5, 1e-4, 1e-3, 1e-2, 1e0, 1e+1, 1e+2\}$. We select the best option between using both data and training labels ($\mathbf{\Gamma} = \mathbf{PDP}$) and using only the data ($\mathbf{\Gamma} = \mathbf{D}$). The label kernel $\mathbf{P}$ is constructed using only the training labels from each data split. Additionally, we choose between edge addition and edge deletion.

For $|k|$, the number of added or deleted edges, we search for the optimal value in $\{0.1m, 0.3m, 0.5m, 0.7m, 0.9m, m\}$, where $m = |\mathcal{E}_r|$ (the number of edges in the reference graph) when adding edges, or $m = |\mathcal{E} \cap \mathcal{E}_r^c|$ (the number of common edges between the original graph and the complement of the reference graph) when deleting edges.

**SDRF hyperparameters.** For each hyperparameter, we search within the range reported in the original paper. Specifically, for the removal bound ($C^+$), we search for the optimal value in $\{0.5, 1, 10, 20, 40\}$; for the $\tau$ parameter, in $\{50, 100, 200\}$; and for the maximum number of itera-

tions, we define $n$ as the number of nodes in each dataset or the cluster size when using our clustering adaptation for large datasets, and search for the optimal value in $\{0.1n, 0.3n, 0.5n, 0.7n, 0.9n, n\}$.

**FoSR hyperparameters.** We search for the optimal value of the maximum number of iterations in $\{50, 100, 150, 200, 300\}$, based on the range reported in the original paper.

**BORF hyperparameters.** For each hyperparameter, we search within the range reported in the original paper. Specifically, for the number of added edges ($h$), we search for the optimal value in $\{20, 30\}$; for the number of deleted edges ($k$), in $\{10, 20, 30\}$; and for the number of batches ($n$), in $\{2, 3\}$.

# E ADDITIONAL RESULTS AND FULL TABLES

## E.1 COMPLETE RESULTS

Table 4: Complete results with standard error of the mean (SEM) for datasets containing up to 5,500 nodes. "T/O" indicates a timeout and "OOM" indicates out of memory. Best results are bolded.

| | Cornell | Texas | Wisconsin | Chameleon | Squirrel | BlogCatalog |
|---|---|---|---|---|---|---|
| Nodes | 183 | 183 | 251 | 851 | 2223 | 5196 |
| $H(\mathcal{G})$ | 0.12 | 0.06 | 0.17 | 0.23 | 0.2 | 0.4 |
| | | | GCN | | | |
| None | $51.8 \pm 1.3$ | $59.7 \pm 2.6$ | $57.2 \pm 1.9$ | $41.3 \pm 0.6$ | $40.7 \pm 0.4$ | $77.6 \pm 0.8$ |
| SDRF | $58.4 \pm 2.2$ | $65.4 \pm 1.8$ | $68.6 \pm 0.8$ | $40.6 \pm 1.0$ | $\mathbf{41.5 \pm 0.6}$ | $77.9 \pm 0.7$ |
| FoSR | $51.6 \pm 2.5$ | $62.4 \pm 1.9$ | $60.5 \pm 1.3$ | $43.1 \pm 1.1$ | $39.7 \pm 0.6$ | $77.4 \pm 0.6$ |
| BORF | $53 \pm 2.7$ | $62.1 \pm 1.8$ | $56.3 \pm 2.3$ | $41.6 \pm 1$ | $40.3 \pm 0.6$ | $78 \pm 0.5$ |
| REFine | $\mathbf{71.3 \pm 1.5}$ | $\mathbf{79.1 \pm 1.6}$ | $\mathbf{82.5 \pm 1.6}$ | $\mathbf{44.1 \pm 1.1}$ | $41.1 \pm 0.7$ | $\mathbf{85.2 \pm 0.3}$ |
| | | | GATv2 | | | |
| None | $43.7 \pm 2.8$ | $53.2 \pm 3.2$ | $53.3 \pm 1.8$ | $40.8 \pm 0.8$ | $37.4 \pm 0.6$ | $80.3 \pm 0.6$ |
| SDRF | $51 \pm 2.4$ | $61.8 \pm 1.3$ | $63.3 \pm 1.3$ | $39.5 \pm 1.4$ | $37.7 \pm 0.6$ | $83.3 \pm 0.9$ |
| FoSR | $46 \pm 2.2$ | $59.7 \pm 1.6$ | $60.9 \pm 1.8$ | $40.1 \pm 0.6$ | $37.7 \pm 0.4$ | $81.6 \pm 1.4$ |
| BORF | $44.6 \pm 2.3$ | $55.1 \pm 2.5$ | $52.5 \pm 2.1$ | $41.2 \pm 1.2$ | $36.7 \pm 0.6$ | $82.2 \pm 1.4$ |
| REFine | $\mathbf{74 \pm 2}$ | $\mathbf{82.4 \pm 2.1}$ | $\mathbf{84.9 \pm 1.3}$ | $43.5 \pm 1.8$ | $\mathbf{38.8 \pm 0.5}$ | $\mathbf{85.9 \pm 1.3}$ |
| | | | APPNP | | | |
| None | $49.4 \pm 1.7$ | $61.9 \pm 2$ | $62.1 \pm 1.3$ | $40.2 \pm 1.1$ | $35.4 \pm 0.7$ | $95.7 \pm 0.3$ |
| SDRF | $63.7 \pm 2.1$ | $77 \pm 1.4$ | $75 \pm 0.9$ | $41 \pm 1.1$ | $35.6 \pm 0.7$ | $95.8 \pm 0.2$ |
| FoSR | $55.1 \pm 1.5$ | $67 \pm 1.4$ | $68.4 \pm 1.8$ | $41.8 \pm 1$ | $35.7 \pm 0.6$ | $\mathbf{95.9 \pm 0.2}$ |
| BORF | $55.1 \pm 2$ | $65.1 \pm 2.4$ | $66 \pm 1.6$ | $39.6 \pm 0.8$ | $36.2 \pm 0.4$ | $95.5 \pm 0.2$ |
| REFine | $\mathbf{74.6 \pm 1.5}$ | $\mathbf{82.4 \pm 1.7}$ | $\mathbf{86 \pm 1.4}$ | $\mathbf{44.5 \pm 1.2}$ | $\mathbf{38.8 \pm 0.8}$ | $95.7 \pm 0.2$ |

Table 5: Complete results with standard error of the mean (SEM) for datasets with more than 5,500 nodes. "T/O" indicates a timeout and "OOM" indicates out of memory. Best results are bolded.

| | Actor | BGP | Tolokers | Roman-empire | Questions | EllipticBitcoin | Genius |
|---|---|---|---|---|---|---|---|
| Nodes | 7600 | 10k | 11k | 22k | 48k | 203k | 421k |
| $H(\mathcal{G})$ | 0.21 | 0.28 | 0.59 | 0.04 | 0.84 | 0.71 | 0.59 |
| | | | | GCN | | | |
| None | $28.4 \pm 0.2$ | $53.4 \pm 0.5$ | $77.2 \pm 0.3$ | $37 \pm 0.3$ | $65.7 \pm 0.4$ | $87.1 \pm 0.05$ | $83.1 \pm 0.07$ |
| SDRF | $29.2 \pm 0.3$ | $53.9 \pm 0.5$ | $77.6 \pm 0.4$ | $46.2 \pm 0.2$ | OOM | $87 \pm 0.04$ | OOM |
| FoSR | $28.1 \pm 0.2$ | $53.3 \pm 0.4$ | $77.4 \pm 0.3$ | $36.9 \pm 0.4$ | $63.3 \pm 0.3$ | $85.9 \pm 0.05$ | $82.2 \pm 0.05$ |
| BORF | $28.3 \pm 0.3$ | $52 \pm 0.8$ | $77 \pm 0.3$ | $35.2 \pm 0.3$ | $65.9 \pm 0.5$ | T/O | T/O |
| REFine | $\mathbf{31.3 \pm 0.4}$ | $\mathbf{59.3 \pm 0.2}$ | $\mathbf{78 \pm 0.2}$ | $\mathbf{58.8 \pm 0.2}$ | $\mathbf{70.3 \pm 0.4}$ | $\mathbf{89.5 \pm 0.08}$ | $\mathbf{83.8 \pm 0.04}$ |
| | | | | GATv2 | | | |
| None | $29.6 \pm 0.4$ | $62.3 \pm 0.3$ | $79.3 \pm 0.2$ | $14.8 \pm 0.4$ | $67.4 \pm 0.5$ | $89.6 \pm 0.4$ | $81.7 \pm 0.1$ |
| SDRF | $29.7 \pm 0.3$ | $63.2 \pm 0.2$ | $\mathbf{79.9 \pm 0.2}$ | $20.8 \pm 0.2$ | OOM | $90.6 \pm 0.1$ | OOM |
| FoSR | $29.2 \pm 0.5$ | $62.8 \pm 0.3$ | $79.5 \pm 0.3$ | $14.7 \pm 0.4$ | $\mathbf{67.6 \pm 0.5}$ | $89.9 \pm 0.1$ | $81.2 \pm 0.06$ |
| BORF | $28.6 \pm 0.5$ | $63 \pm 0.3$ | $79.4 \pm 0.2$ | $14.9 \pm 0.4$ | $\mathbf{67.6 \pm 0.4}$ | T/O | T/O |
| REFine | $\mathbf{35.1 \pm 0.3}$ | $\mathbf{63.3 \pm 0.3}$ | $79.7 \pm 0.3$ | $\mathbf{28.5 \pm 0.7}$ | $66.6 \pm 0.5$ | $\mathbf{90.8 \pm 0.05}$ | $\mathbf{83.6 \pm 0.03}$ |
| | | | | APPNP | | | |
| None | $33.8 \pm 0.2$ | $63.6 \pm 0.5$ | $71.1 \pm 0.3$ | $14 \pm 0.07$ | $44.1 \pm 3.7$ | $87.4 \pm 0.05$ | $81.9 \pm 0.4$ |
| SDRF | $33.8 \pm 0.2$ | $63.6 \pm 0.3$ | $71.8 \pm 0.2$ | $22.6 \pm 0.06$ | OOM | $87.4 \pm 0.03$ | OOM |
| FoSR | $33.9 \pm 0.2$ | $63.6 \pm 0.5$ | $71.9 \pm 0.3$ | $14.3 \pm 0.4$ | $44.8 \pm 3.5$ | $86.7 \pm 0.05$ | $81.2 \pm 0.4$ |
| BORF | $33.6 \pm 0.3$ | $63.4 \pm 0.3$ | $71.4 \pm 0.2$ | $15.5 \pm 0.6$ | $44.5 \pm 3.2$ | T/O | T/O |
| REFine | $\mathbf{34.8 \pm 0.3}$ | $\mathbf{64.3 \pm 0.3}$ | $\mathbf{73.8 \pm 0.2}$ | $\mathbf{30.8 \pm 0.5}$ | $\mathbf{47 \pm 2.6}$ | $\mathbf{89.8 \pm 0.09}$ | $\mathbf{83.6 \pm 0.04}$ |

Table 6: Complete GCN results with standard error of the mean (SEM) for high-homophily datasets. Best results are bolded.

|  | Cora | Citeseer | Pubmed |
|---|---|---|---|
| Nodes | 2708 | 3327 | 19K |
| $H(\mathcal{G})$ | 0.8 | 0.73 | 0.8 |
| None | $\mathbf{87.6 \pm 0.5}$ | $\mathbf{77.3 \pm 0.3}$ | $88.2 \pm 0.2$ |
| SDRF | $\mathbf{87.6 \pm 0.5}$ | $77.2 \pm 0.5$ | $\mathbf{88.3 \pm 0.2}$ |
| FoSR | $87 \pm 0.6$ | $76.7 \pm 0.4$ | $88.2 \pm 0.2$ |
| BORF | $86.6 \pm 0.6$ | $76.6 \pm 0.4$ | $87.3 \pm 0.2$ |
| REFine | $87.4 \pm 0.4$ | $77.2 \pm 0.4$ | $\mathbf{88.3 \pm 0.2}$ |

Table 7: Complete results with standard error of the mean (SEM) on heterophilic graphs for specialized GNNs vs. ST+REFine. "OOM" indicates out-of-memory. The best results are bolded, and the second-best are underlined.

|  | Cornell | Texas | Wisconsin | Chameleon | Squirrel | BlogCatalog | Actor | BGP | Roman-empire |
|---|---|---|---|---|---|---|---|---|---|
| MixHop | $71.9 \pm 1.5$ | $79.1 \pm 1.7$ | $83.1 \pm 1.7$ | $43.2 \pm 1.3$ | $39.8 \pm 0.7$ | OOM | $\mathbf{36.2 \pm 0.3}$ | $64.3 \pm 0.2$ | $32.1 \pm 1.5$ |
| H$_2$GCN | $73.2 \pm 1.8$ | $\mathbf{82.7 \pm 2}$ | $82.3 \pm 1.6$ | $41.8 \pm 1$ | $40.4 \pm 0.4$ | $\mathbf{96.4 \pm 0.1}$ | $30.3 \pm 0.5$ | $64.9 \pm 0.5$ | $34.3 \pm 1.7$ |
| GPRGNN | $70.8 \pm 1.2$ | $81 \pm 1.7$ | $82.5 \pm 1$ | $40.9 \pm 1$ | $38.5 \pm 0.8$ | $95.7 \pm 0.1$ | $35.4 \pm 0.3$ | $\underline{65 \pm 0.5}$ | $20.5 \pm 3.6$ |
| OrderedGNN | $70.8 \pm 2.1$ | $77.8 \pm 1.4$ | $82.1 \pm 1.1$ | $38 \pm 1.1$ | $34.3 \pm 0.7$ | $\underline{95.7 \pm 0.2}$ | $35.8 \pm 0.3$ | $\mathbf{65 \pm 0.1}$ | $45.5 \pm 0.6$ |
| ST+REFine | $\mathbf{74.6 \pm 1.5}$ | $82.4 \pm 1.7$ | $\mathbf{86 \pm 1.4}$ | $\mathbf{44.5 \pm 1.2}$ | $\mathbf{41.1 \pm 0.7}$ | $\underline{95.7 \pm 0.2}$ | $35.1 \pm 0.3$ | $64.3 \pm 0.3$ | $\mathbf{58.8 \pm 0.2}$ |

## E.2 IMPACT OF REWIRING ON EDGE HOMOPHILY

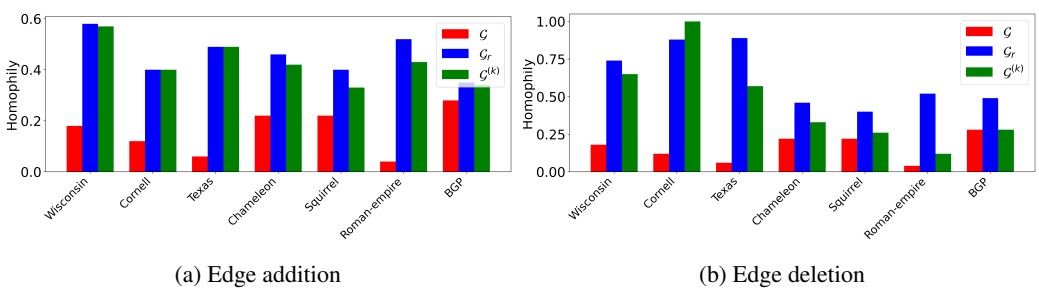

(a) Edge addition

(b) Edge deletion

Figure 8: Edge homophily of the original graph $\mathcal{G}$, the reference graph $\mathcal{G}_r$ used for rewiring, and the rewired graph $\mathcal{G}^{(k)}$.

## E.3 REFERENCE GRAPH HOMOPHILY: FEATURES VS. LABEL-DRIVEN DIFFUSION

Table 8 compares the homophily of the reference graph $H(\mathcal{G}_r)$ when constructed using only node features ($\mathbf{\Gamma = D}$), using label-driven diffusion that incorporates both features and training labels ($\mathbf{\Gamma = PDP}$), and the baseline homophily of the original graph $H(\mathcal{G})$. As shown, both reference graphs exhibit consistently higher homophily than the original graph. Moreover, incorporating label information ($\mathbf{\Gamma = PDP}$) further improves homophily in most cases compared to using features alone ($\mathbf{\Gamma = D}$).

Table 8: Homophily of the reference graph $\mathcal{G}_r$ constructed using only node features ($\boldsymbol{\Gamma} = \mathbf{D}$), using label-driven diffusion ($\boldsymbol{\Gamma} = \mathbf{PDP}$), and the original graph $\mathcal{G}$.

| | Cornell | Texas | Wisconsin | Chameleon | Squirrel | BlogCatalog | Actor | BGP | Roman-empire |
|---|---|---|---|---|---|---|---|---|---|
| $H(\mathcal{G})$ | 0.12 | 0.06 | 0.17 | 0.23 | 0.2 | 0.4 | 0.21 | 0.28 | 0.04 |
| $H(\mathcal{G}_r)$ ($\mathbf{D}$) | 0.47 | 0.55 | 0.58 | **0.28** | **0.29** | 0.4 | 0.24 | 0.4 | **0.52** |
| $H(\mathcal{G}_r)$ ($\mathbf{PDP}$) | **0.66** | **0.7** | **0.7** | 0.25 | 0.25 | **0.65** | **0.27** | **0.6** | 0.44 |

## F  COMPLEXITY AND RUNTIME

The computation of the affinity kernels $\mathbf{D}$ and $\mathbf{P}$ has a time complexity of $O(d \cdot c^2)$, where $d$ is the dimension of the node feature vectors and $c$ is the cluster size, which we set to 100 or 500 in our experiments. The multiplication of the kernels to obtain $\boldsymbol{\Gamma}$ incurs a complexity of $O(c^3)$. The total number of clusters is given by $\frac{n}{c}$, where $n$ denotes the total number of nodes in the graph. Since the computational complexity per cluster is $O(c^3)$, the overall complexity is $O(c^3 \cdot \frac{n}{c}) = O(c^2 \cdot n)$, neglecting clustering via METIS (average-case $O(|\mathcal{E}|)$), which is negligible on standard sparse benchmarks ($|\mathcal{E}| = O(n)$). Notably, when $n \gg c$, the complexity simplifies to $O(n)$, indicating that the method remains linear in the number of nodes.

### F.1  PARALLEL IMPLEMENTATION ON GPU

The bottleneck in our method is matrix multiplication. Since METIS partitions the graph into clusters of approximately equal size, it enables us to implement the matrix multiplication in parallel on the GPU. Thus, given $g$ units of GPU, the overall complexity is $O(c^3 \cdot \frac{n}{c} \cdot \frac{1}{g}) = O(\frac{c^2}{g} \cdot n)$, making our method even more efficient in practice compared to other approaches.

### F.2  COMPLEXITY COMPARISONS

Table 9: Comparison of method complexities: $n$ nodes, $c$ cluster size, $m$ edges, and $d_{\max}$ max node degree.

| Method | Complexity |
|---|---|
| SDRF | $O(md_{\max}^2)$ per edge |
| BORF | $O(md_{\max}^3)$ per cluster |
| FoSR | $O(n^2)$ per edge |
| REFine (Ours) | $O(c^3)$ per cluster |

### F.3  RUNTIME COMPARISONS

In Table 10, we compare the runtimes of our REFine method with baseline approaches across three datasets. Due to the high computational cost of the baselines on large datasets, we adapt them to use the same clustering strategy as REFine (described in Section 5). For small datasets (with fewer than 1000 nodes), we use the original implementations. For larger datasets (with 1000 or more nodes), we apply our clustering-based adaptations, using a cluster size of 500 for BlogCatalog and Roman-empire. For the runtime comparison, we use default hyperparameters for all methods. Specifically, for SDRF, we set $C^+ = 0.5$, $\tau = 100$, and the maximum number of iterations to $0.2n$, where $n$ is the number of nodes in the dataset (or the cluster size when applying the clustering adaptation). For FoSR, the maximum number of iterations is 50 for each cluster. For BORF, we set $h = 30$, $k = 20$, and $n = 3$.

Table 10: Runtime comparison across different methods. The table shows the runtimes (in seconds) for SDRF, FoSR, BORF, and our REFine method on three datasets: Wisconsin, BlogCatalog, and Roman-empire. $n$ represents the number of nodes and $m$ represents the number of edges. The smallest runtime for each dataset is highlighted in bold.

| Dataset | | | SDRF | FoSR | BORF | REFine (ours) |
|---|---|---|---|---|---|---|
| Name | $n$ | $m$ | | | | |
| Wisconsin | 251 | 900 | 0.8 | 4.7 | 3.8 | **0.1** |
| BlogCatalog | 5196 | $343k$ | 29 | 5.7 | 180 | **3.3** |
| Roman-empire | $22k$ | $65k$ | 20 | 5.8 | 380 | **4.2** |

## F.4  NEGLIGIBLE OVERHEAD OF METIS CLUSTERING

REFine has per-cluster complexity $\mathcal{O}(c^3)$, and with $n/c$ clusters this yields $\mathcal{O}(c^2 n)$, where $n$ is the number of nodes and $c$ is the cluster size. Including clustering with METIS (average-case $\mathcal{O}(|\mathcal{E}|)$), the end-to-end complexity is $\mathcal{O}(|\mathcal{E}| + c^2 n)$. On standard sparse benchmarks ($|\mathcal{E}| = \mathcal{O}(n)$), when $c^2 n \gg |\mathcal{E}|$ the complexity reduces to $\mathcal{O}(c^2 n)$ and the $|\mathcal{E}|$ term is negligible.

For instance, in the Genius dataset ($n = 421k$, $|\mathcal{E}| = 989k$, $c = 100$), $c^2 n \gg |\mathcal{E}|$, and METIS runtime is negligible. The table below reports runtime (in seconds) for METIS, REFine, and competing methods using the same clustering, showing that METIS adds negligible overhead:

Table 11: Runtime (in seconds) of METIS clustering, REFine, and competing methods. Results on BGP and Roman-Empire show that METIS adds negligible overhead compared to the total runtime.

| Dataset | nodes | edges | METIS(clustering) | SDRF | FoSR | BORF | REFine(ours) |
|---|---|---|---|---|---|---|---|
| BGP | 10k | 206k | 0.09 | 84 | 5.79 | 95 | 1.75 |
| Roman-empire | 22k | 65k | 0.03 | 20 | 5.8 | 380 | 4.2 |

# G  ABLATION STUDIES & ANALYSIS

## G.1  HOMOPHILY AND REWIRING EFFECTIVENESS

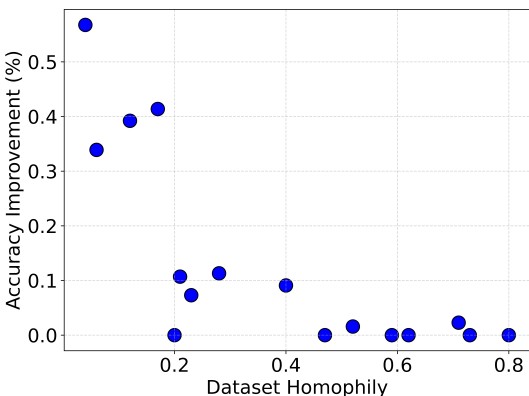

Figure 9: Test accuracy improvement across all evaluated datasets.

## G.2  REWIRING ONLY USING TRAIN LABELS

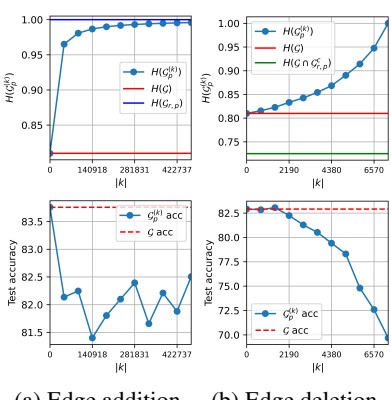

(a) Edge addition    (b) Edge deletion

Figure 10: Rewiring Cora using $\mathcal{G}_r$ built only from train labels.

## G.3  CLUSTER SIZE

In our experiments, we set the default cluster size to 500 for datasets with $1000 < n \leq 25000$ and 100 for datasets with $n > 25000$, primarily to optimize runtime. Table 12 presents the effect of varying the cluster size $\{100, 500, 1000, 2000\}$ when applying REFine. The 'Improvement' column indicates whether changing the default cluster size leads to a statistically significant improvement. As shown, while increasing the cluster size beyond 500 can sometimes yield improvements, the overall effect remains relatively minor.

Table 12: Effect of varying the cluster size on REFine performance across different datasets. We report the mean test accuracy with the standard error of the mean (SEM) for different architectures (GCN, GATv2, APPNP). The 'Improvement' column indicates whether increasing the cluster size results in a statistically significant improvement (V) or not (X).

| Dataset | $n$ | $H(\mathcal{G})$ | Arch | 100 | 500 | 1000 | 2000 | Improvement |
|---------|-----|------|------|-----|-----|------|------|-------------|
| Squirrel | 2223 | 0.2 | GCN | $40.5 \pm 0.8$ | $41.1 \pm 0.7$ | $40.7 \pm 0.6$ | N/A | X |
| | | | GATv2 | $37.4 \pm 0.6$ | $38.8 \pm 0.5$ | $38.9 \pm 0.5$ | N/A | X |
| | | | APPNP | $37.2 \pm 0.6$ | $38.8 \pm 0.8$ | $38.8 \pm 0.7$ | N/A | X |
| BlogCatalog | 5196 | 0.4 | GCN | $80.9 \pm 0.5$ | $85.2 \pm 0.3$ | $86.2 \pm 0.5$ | $86.4 \pm 0.3$ | V |
| | | | GATv2 | $80.7 \pm 0.7$ | $85.9 \pm 1.3$ | $83.1 \pm 1.2$ | $82.4 \pm 2.1$ | X |
| | | | APPNP | $95.9 \pm 0.3$ | $95.7 \pm 0.2$ | $95.3 \pm 0.2$ | $94.8 \pm 0.2$ | X |
| Roman-empire | 22k | 0.04 | GCN | $57 \pm 0.2$ | $58.8 \pm 0.2$ | $59.5 \pm 0.2$ | $59.7 \pm 0.2$ | V |
| | | | GATv2 | $27.4 \pm 0.8$ | $28.5 \pm 0.7$ | $28.5 \pm 1.1$ | $29.7 \pm 2$ | X |
| | | | APPNP | $29.7 \pm 0.7$ | $30.8 \pm 0.5$ | $29.2 \pm 0.8$ | $30.7 \pm 0.5$ | X |
| EllipticBitcoin | 203k | 0.71 | GCN | $89.5 \pm 0.08$ | $89.7 \pm 0.07$ | $89.8 \pm 0.07$ | $90 \pm 0.06$ | V |
| | | | GATv2 | $90.8 \pm 0.05$ | $90.8 \pm 0.07$ | $90.7 \pm 0.15$ | $90.7 \pm 0.1$ | X |
| | | | APPNP | $89.8 \pm 0.09$ | $90.1 \pm 0.05$ | $90.3 \pm 0.03$ | $90.3 \pm 0.09$ | V |

## H  USE OF LLMS

In this paper, LLMs were employed solely as an aid to improve the clarity and readability of the text. Their role was limited to assisting with polishing the writing style and grammar, without influencing the research process, methodology, or results.

