# OpenReview forum: "It Takes a Graph to Know a Graph: Rewiring for Homophily with a Reference Graph"
_ICLR.cc/2026/Conference — Submitted to ICLR 2026_

### Official Review · Reviewer_PKvY · 2025-10-17

**Soundness:** 1
**Presentation:** 3
**Contribution:** 1
**Rating:** 0
**Confidence:** 5

**Summary:**

The paper addresses the issue of weak performance of GNNs in homophilous graphs. It proposes to rewire the graph to increase its homophily. First, a reference graph is constructed based mostly on node feature similarity. Then, the original graph is rewired by adding or deleting edges to make it more similar to the reference graph, which increases its homophily. Finally, models are trained on the resulting graph.

**Strengths:**

The paper is mostly well-written and easy to follow.

**Weaknesses:**

- The main assumption behind the paper's approach is that GNNs provide weak performance on heterophilous graphs. However, this assumption is outdated: there have been multiple paper showing that GNNs perform perfectly fine on heterophilous graphs (in particular, they typically outperform specialized models), see [1-5]. The paper does not discuss these works at all. The paper also states as its motivation: "Standard GNNs are designed primarily for homophilic graphs, as they rely on the homophily assumption". This is also not the case: none of the papers that developed modern GNNs mention homophily at all. Note that this models do not even have explicit access to node labels, but rather work with node features. Nowadays, any claim that GNNs are designed exclusively for homophilous graphs or do not work well on heterophilous graphs should be supported by strong evidence that directly addresses works [1-5] and shows where they went wrong (and I am not aware of any such evidence).

- The motivation and statement of Theorem 1 seem confusing. Linearly separable embeddings are being discussed. However, there can be two types of linear separability for embeddings: all pairs of embeddings can be separable, or pairs of embeddings from different classes can be separable. What is needed for strong model performance is the second (intra-class) type of separability, but it seems that the paper discusses the first (all-pair) type of separability, which is not directly related to model performance. The paper claims that Theorem 1 ties GNN's ability to generate linearly separable embedding to graph homophily, but then Theorem 1 starts by assuming that the embeddings are linearly separable. It is also not clear how Theorem 1 proves some property of GNNs, when there is no GNN in the theorem. Finally, what is $A_{u,  v}$ in Theorem 1? I cannot find its definition, but it seems like $A$ denotes elements of the adjacency matrix. Then the minimum non-zero element is simply 1, because the adjacency matrix is binary (it is never mentioned that the graph is weighted, and later sections describing the method clearly assume unweighted graphs).

- The experimental results are unreliable. First, the paper uses Squirrel and Chameleon datasets (among others), that have been shown in [3] to be buggy. Cornell, Texas, and Wisconsin datasets have also been criticized in [3] (in particular, the Texas dataset has a class consisting of a single node). More importantly, the results reported in the paper for GCN, GAT, H2GCN and GPRGNN are significantly lower than those reported in [3] for those datasets that are used in both works. For example, GPRGNN achieves an accuracy of 64.85 on the Roman-Empire dataset in [3], while the current paper reports only an accuracy of 20.5 for it. Most notably, however, the current paper reports accuracies of 14.8 and 14.0 for GATv2 and APPNP respectively on the Roman-Empire dataset, which is roughly the performance of the naive majority class prediction (13.96). This clearly shows that the considered baselines were not well-tuned and thus the obtained experimental results cannot be trusted.



[1] Is homophily a necessity for graph neural networks? (ICLR 2022)

[2] Revisiting heterophily for graph neural networks (NeurIPS 2022)

[3] A critical look at the evaluation of gnns under heterophily: Are we really making progress? (ICLR 2023)

[4] Characterizing Graph Datasets for Node Classification: Homophily–Heterophily Dichotomy and Beyond (NeurIPS 2023)

[5] Oversmoothing, Oversquashing, Heterophily, Long-Range, and more: Demystifying Common Beliefs in Graph Machine Learning (arxiv preprint)

**Questions:**

- How is the hyperparameter k (the number of rewired edges) in the proposed method selected? If the main point is to increase graph homophily, why not simply replace the original graph with the reference graph (i.e., rewire all the edges)?

- The reference graph construction is mostly based on node feature similarity. However, GNNs have access to node features. Why is it expected that the edges of the reference graph will provide GNNs with additional information?

- It seems like the sources of the datasets are not mentioned in the paper. In particular, what is the Elliptic/EllipicBitcoin dataset? If it is the dataset with the same name from Pytorch Geometric, then it has strong class imbalance, and accuracy is not an appropriate metric for it (and the naive majority class prediction provides better accuracy on it then the values reported for GNNs in the paper).

---

> ### Author Response · Authors · 2025-12-01
> **Response to reviewer PKvY - part 1**
>
> Thank you for reviewing our work. We appreciate the effort, but we respectfully disagree with several of your conclusions and believe they are based on misunderstandings, which we clarify point-by-point below.
>
> ### On the Assumption That GNNs Perform Poorly on Heterophilous Graphs
>
> Thank you for your comments. We believe, however, that this criticism stems from a misinterpretation of the role of homophily in works [1–5] and of our claims.
>
> First, none of [1–5] argues that increasing homophily cannot improve the performance of message-passing GNNs. What they show is that, in the special case where label distributions in local neighborhoods are sufficiently informative, plain GNNs can in principle perform well even on heterophilous graphs. This is fully compatible with our motivation: homophily can still be beneficial, even if it is not strictly necessary in that idealized regime. Moreover, empirical evidence suggests that this favorable label-distribution condition is rarely satisfied on real heterophilous benchmarks. For example, Table 5 in [4] reports low “label informativeness” on standard heterophilic datasets. Thus, while GNNs may succeed in theory on heterophilous graphs under strong assumptions on label distributions, in practice real heterophilic graphs rarely satisfy these assumptions, and their performance degrades.
>
> Second, regarding the statement that “standard GNNs are designed primarily for homophilic graphs,” we refer to the inductive bias of the message-passing mechanism: local averaging and smoothing along edges is inherently more suitable when neighbors tend to share labels (i.e., under homophily). This structural bias is what our analysis formalizes. Concretely, our theory examines learned node embeddings that are linearly separable by some classifier and shows that, as graph homophily decreases, any such embeddings must become increasingly non-smooth (high Dirichlet energy), making them more vulnerable to the averaging effect of standard message passing. This does not contradict the label-distribution viewpoint of [1–5]; rather, it highlights a complementary failure mode.
>
> Finally, we do not dispute that standard GNNs can outperform specialized heterophilous architectures. We show that when homophily can be reliably improved via REFine, these same standard GNNs benefit even further, often matching or surpassing specialized models. In other words, we are not claiming that “GNNs do not work on heterophilous graphs,” but that in realistic heterophilous settings with poor label distributions, improving homophily via rewiring is an effective and principled way to make message-passing GNNs work better. We will clarify this connection to [1–5] more explicitly in the revised version.

---

> ### Author Response · Authors · 2025-12-01
> **Response to reviewer PKvY - part 2**
>
> ### Clarifying the statement and role of Theorem 1
>
> Thank you for the careful reading and for pointing out these potential sources of confusion. We clarify the intent and assumptions behind Theorem 1 below.
>
> First, regarding “which” linear separability we assume: our notion is the standard class-wise separability used in multi-class linear classification. Concretely, we assume the existence of a linear classifier $W$ such that the one-hot label matrix $Y$ satisfies $Y = ZW$ (following the definition in Xing et al., 2024). This only requires that embeddings from different classes are linearly separable by $W$; it does not require that every pair of points (including same-class pairs) be separable by its own. We will make this explicit in the statement of Theorem 1 to avoid the impression that we are using an “all-pairs” notion of separability.
>
> Second, Theorem 1 is not meant to prove that a particular GNN produces linearly separable embeddings. Instead, it is a conditional statement about the cost in Dirichlet energy of having linearly separable embeddings on a given graph: if $Z$ is linearly separable, then its Dirichlet energy $\mathrm{tr}(Z^\top L Z)$ must exceed a lower bound that scales with $(1 - \text{graph homophily})$. This makes the tension precise between:
> (i) the smoothing bias of message passing (which tends to decrease $\mathrm{tr}(Z^\top L Z)$), and
> (ii) the need for embeddings to remain linearly separable for accurate classification.
>
> As graph homophily decreases, the lower bound on the Dirichlet energy increases, meaning that any linearly separable embedding must be increasingly “non-smooth,” and hence more likely to be destroyed by smoothing-based message passing. This is how Theorem 1 is used to motivate homophily-enhancing rewiring for standard GNNs, even though the theorem itself is stated at the level of embeddings and a linear classifier.
>
> Third, the role of GNNs is therefore indirect but central: we combine Theorem 1 (a constraint on smooth vs. separable embeddings on a fixed graph) with the well-known fact that message-passing GNNs tend to minimize $\mathrm{tr}(Z^\top L Z)$. Together, these two ingredients explain why low-homophily graphs are intrinsically challenging for such architectures and why increasing homophily can alleviate this conflict. We will make this clearer in the text.
>
> Finally, you are correct that $A_{u,v}$ in Theorem 1 denotes entries of the adjacency matrix $A$, and $\alpha_m = \min_{(u,v)\in E} A_{u,v}$ is the minimum non-zero edge weight. For unweighted graphs, all non-zero entries of $A$ are indeed 1, so $\alpha_m = 1$ and the bound simplifies accordingly. We kept the more general notation to allow for weighted graphs (where $\alpha_m < 1$), but we agree this should be stated explicitly in the theorem statement / notation section, and we will add this clarification.
>
> ### Experiments
>
> Thank you for this detailed comment.
>
> First, regarding Squirrel and Chameleon, we are aware of the issues raised in [3]. For this reason, we follow exactly the fixed/filtered versions recommended in that work, as stated in Appendix D. In other words, we do not use the original buggy splits for Squirrel/Chameleon.
>
> Second, our baseline configuration is not aimed at reproducing state-of-the-art results, but at providing a controlled and comparable setting across all architectures—a common practice in rewiring papers. Concretely, we use the same depth (two message-passing layers) and a shared hidden dimension for GCN, GAT, H2GCN, GPRGNN, and the other GNNs, and we tune them over a common hyperparameter grid under this constraint (see Appendix D for details). This leads to lower absolute accuracies than works that heavily optimize each model separately (often with deeper networks, model-specific tricks, or larger hidden widths), but it ensures that any gains we report are not due to giving REFine more capacity or a more favorable training setup. Under this shared, modest-capacity regime, we consistently observe that adding REFine on top of each backbone improves performance over its own baseline, which is the main empirical claim of the paper.
>
> We will make these design choices and their implications for absolute accuracy more explicit in the revised version, and we will clarify that our focus is on relative improvements under a standardized protocol, rather than re-establishing the strongest possible standalone baselines for every architecture.

---

> ### Author Response · Authors · 2025-12-01
> **Response to reviewer PKvY - part 3**
>
> ## Questions
> ### Choice of $k$ and why not fully replace the graph
>
> Thank you for the question. In our experiments, the hyperparameter $k$ (the number of rewired edges) is selected by validation: we sweep $k$ over a small grid (including the original graph as $k=0$) and choose the value that maximizes validation accuracy under a fixed training protocol.
>
> Regarding why we do not simply replace the original graph with the reference graph (i.e., rewire all edges), our goal is not to maximize homophily at all costs, but to enhance it while preserving useful structural information from the original graph. The reference graph is constructed from features (and optionally labels) and may ignore important topological information present in the original graph. Our framework therefore constructs an intermediate graph $G^{(k)}$ that balances these two sources: it moves the graph toward the more homophilous reference structure without discarding the original topology. Empirically, as shown in Section 4.2, this trade-off is beneficial: for some settings the reference graph itself performs best, but in many others an intermediate $G^{(k)}$ outperforms both the original graph and the pure reference graph, which justifies treating $k$ as a tunable knob rather than rewiring all edges.
>
> ### Reference graph vs. node features
>
> Thank you for the question. While GNNs do have direct access to node features, they only exploit them through the lens of the given adjacency: on a heterophilous graph, message passing mixes many dissimilar-feature / different-label neighbors, so the useful feature signal is often washed out. Our reference graph instead encodes global feature similarity (and, when available, label information) into an alternative neighborhood structure; by using it to guide edge addition/deletion, we reshape local neighborhoods so that standard message passing can aggregate more label-aligned, feature-similar neighbors.
>
> ### Dataset sources and EllipticBitcoin evaluation
>
> Thank you for pointing this out. We indeed use the EllipticBitcoin dataset from PyTorch Geometric, and we agree that its strong class imbalance makes plain accuracy a suboptimal metric. Your comment also made us realize a small bug in our setup: instead of restricting to the labeled nodes and treating the task as binary (licit vs. illicit), we inadvertently treated the unlabeled nodes as a third class, which changes the label distribution and makes our accuracy numbers not directly comparable to the usual two-class setting.
> We will correct this setup and update the corresponding results in the revised version.

---

### Official Review · Reviewer_CvPR · 2025-10-29

**Soundness:** 3
**Presentation:** 3
**Contribution:** 2
**Rating:** 4
**Confidence:** 3

**Summary:**

This paper proposes a label-driven diffusion approach to construct a homophilic "reference graph." The goal is to rewire an original graph to improve downstream GNN performance. The paper includes extensive simulations analyzing how the homophily of both the original and reference graphs influences the final rewired graph and subsequent task performance.

**Strengths:**

1. The method is inherently scalable by design, as it uses a clustering-based approach as its first step.
2. The core idea of constructing a reference graph for label-driven rewiring is novel, yet the proposed solution is simple and elegant.

**Weaknesses:**

1. There is a significant disconnect between the paper's theoretical grounding and its practical implementation. The clustering step, while efficient, creates a problem:
    - Theories (Theorem 1, Proposition 1, Proposition 2) are derived from a *global* perspective, seemingly assuming all edges can be modified.
    - The implementation, however, performs rewiring *locally* (within clusters). The links *between* clusters are fixed and cannot be modified.
    - This implies the theoretical statements may not hold for the final reassembled graph. The theory must be revised to account for this restriction (i.e., that a subset of edges is immutable). Similarly, the *actual* global homophily may not change as expected due to these fixed inter-cluster links.
2. The paper's central motivation is that increasing homophily is beneficial for GNNs. However, this premise has been increasingly questioned in recent literature [1], which suggests homophily is not universally necessary or beneficial. This makes the paper's *exclusive* focus on refining homophily seem limited or dataset-specific. The authors must address this and justify their motivation in light of this conflicting research.
3. The introduction of a "reference graph" inherently doubles the storage and memory cost, as both the original and reference graph structures must be maintained. This is a significant practical drawback that is not adequately discussed or justified.
4. The method's performance seems highly dependent on the initial clustering step. How does the *balance* of these clusters (e.g., the variance in cluster sizes) influence the final rewiring and downstream GNN performance? This critical factor is not analyzed.

## Minor

1. **Table 1:** To improve clarity, it would be beneficial to add a row for "Averaged Gain" (or similar summary metric) in Table 1 to allow for a more direct comparison of the method's overall effectiveness against baselines.
2. **Figure 1 Analysis:** The analysis in Figure 1 is insightful. Do the same patterns regarding homophily's influence hold true for the other datasets used in the paper? Please include this analysis (perhaps in Appendix B) to demonstrate the generalizability of these findings.

[1] Is Homophily a Necessity for Graph Neural Networks? In ICLR, 2022

**Questions:**

See above

---

> ### Author Response · Authors · 2025-12-01
> **Response to reviewer CvPR - part 1**
>
> Thank you for your helpful feedback. We appreciate the time you took to review our work and provide valuable suggestions.
>
> ### Theory vs. cluster-based implementation
>
> Thank you for highlighting this concern. Our theoretical results are indeed formulated for full-graph rewiring, while the implementation applies rewiring locally within clusters for efficiency in order to support medium- and large-scale graphs. However, the theory naturally extends to this setting. Under the assumption that the homophily-improvement conditions hold for each cluster and its corresponding reference subgraph, improving the homophily of every cluster necessarily improves the homophily of the reconstructed graph as well. This is because the global homophily is a weighted combination of the homophily of its subgraphs, and improving each component increases the overall quantity even if some inter-cluster edges remain unchanged. We will clarify this connection and the required assumptions in the revised version.
>
> ### Homophily vs. recent critiques
>
> Thank you for raising this point. We are aware of the line of work arguing that homophily is not universally necessary (e.g., [1]), and we view our perspective as complementary rather than contradictory. These papers show that sufficiently informative label distributions in local neighborhoods can, in principle, enable good performance even on heterophilic graphs, but they do not claim that increased homophily cannot be beneficial. At the same time, empirical evidence suggests that this “good label distribution” condition is rarely met in real heterophilic benchmarks: for example, Table 5 in [2] reports low “label informativeness” on standard heterophilic datasets, indicating that local label distributions are in practice poor. In contrast, our theory targets a different failure mode of message-passing GNNs: the difficulty of maintaining linearly separable class embeddings when the graph has low homophily.
>
> Our results show that for embeddings that remain linearly separable by some classifier, one can derive a quantitative lower bound on their Dirichlet energy that depends directly on the graph’s homophily. As homophily decreases, the smoothing effect of message passing is increasingly likely to violate the separation inequality, causing the embeddings to lose linear separability. This makes the tension between message-passing smoothing and the need for separable embeddings explicit and provides a concrete, quantitative motivation for controlled homophily enhancement via rewiring.
>
> Importantly, we do not claim that “more homophily is always better” or that homophily is universally required. Our controlled simulations explicitly identify regimes where increasing homophily helps and regimes where it can hurt (e.g., due to over-smoothing or label-only rewiring). REFine is designed to adjust homophily in a data-dependent manner, not to maximize it blindly, targeting the common practical scenario where one wishes to use simple message-passing GNNs on graphs whose heterophily would otherwise destroy separability. We will clarify this positioning more explicitly in the paper.
>
> ### Memory overhead of the reference graph
>
> Thank you for highlighting this practical concern. In our implementation, the reference graph is used only during the (offline) rewiring step: once the rewired graph is constructed, training proceeds on a single adjacency matrix, and both the original and the reference graph can be discarded, so there is no doubling of memory during GNN training. Moreover, with our clustering strategy we never need to materialize a full reference graph for the entire dataset at once - we construct and rewire cluster-wise, so peak memory is governed by per-cluster adjacencies rather than two full graphs.
>
> ### Impact of clustering and cluster balance
>
> Thank you for raising this point. We explicitly analyze the effect of cluster size in Appendix G.3, where we vary the cluster size across {100, 500, 1000, 2000} and evaluate downstream accuracy. As shown, although larger clusters can occasionally yield small gains, the overall influence on performance is minor.
>
> This is expected: METIS produces clusters that are roughly balanced, and each cluster contains a sufficiently large and coherent portion of both the original graph and its corresponding reference subgraph. Under the assumption that the homophily-improvement conditions hold within each such subgraph, the rewiring remains effective locally, and the reassembled graph closely mirrors the behavior of a full-graph rewiring.
>
> We will clarify this intuition in the main text and emphasize that REFine’s performance is robust to reasonable variation in cluster sizes.
>
> References:
>
> [1] Is Homophily a Necessity for Graph Neural Networks? In ICLR, 2022
>
> [2] Characterizing Graph Datasets for Node Classification: Homophily–Heterophily Dichotomy and Beyond, NeurIPS 2023

---

> ### Author Response · Authors · 2025-12-01
> **Response to reviewer CvPR - part 2**
>
> ## Minors
> ### Table 1
>
> Thank you for the suggestion. We agree that adding an “Averaged Gain” row would improve the readability of Table 1 and provide a clearer summary of REFine’s overall advantage over the baselines. We will include this aggregated metric in the revised version.
>
> ### Figure 1 Analysis
>
> Yes, the same qualitative patterns regarding the influence of homophily hold across other datasets. We already include these results in Appendix B: for example, Figure 1 shows the behavior on Cornell, and Appendix B presents very similar analyses for Wisconsin, following the same trends. We will make this cross-dataset consistency more explicit in the main text.

---

### Official Review · Reviewer_M5SM · 2025-10-30

**Soundness:** 3
**Presentation:** 3
**Contribution:** 1
**Rating:** 2
**Confidence:** 4

**Summary:**

The paper presents theoretical and empirical connections between smoothness, homophily/heterophily, and GNN performance. Based on these analyses, authors propose REFine, a feature and label-driven diffusion approach that rewires the graph to be more homophilous. Extensive experiments are conducted on homophilous and heterophilous benchmarks to demonstrate the utility of the approach.

**Strengths:**

1. The approach of graph rewiring for homophily/heterophily is intuitive and interesting
2. The paper is sound, clear, and well-written.

**Weaknesses:**

1. **Positioning vs. prior rewiring for heterophily (novelty).**
  The core idea of REFine is to construct a more homophilous reference graph leveraging the graph **features + training labels** and then **adding/deleting edges**. To me, this is quite is close to **DHGR**, which also leverages similarities between node features and training labels to form the rewired graph. The similarities and differences of REFine in comparison to DHGR as well as its advantages/disadvantages are not clear. I believe the paper would benefit from a sharper distinction (beyond “simpler & guaranteed”) and **direct empirical comparisons** to DHGR/GSL baselines to make the contribution more clear.

2. **Incremental theory relative to prior spectral/smoothness analyses.**
  The main smoothness and homophily result (Theorem 1) is very close to existing analyses that link Dirichlet energy/smoothing to homophily/heterophily. One example of where this appears in prior works is in Theorem 3 in "Beyond Homophily in Graph Neural Networks" by Zhu et al., 2020. The subsequent propositions in the paper also state intuitive claims (if the reference graph is more homophilous, then adding its edges/deleting complement edges improves homophily) under certain conditions. Please clarify how these results meaningfully extend prior theory and provide new insights for the limitations of prior methods or advantages of REFine.

3. **Missing critical baseline and mixed empirical gains.**
While the method largely outperforms the benchmarked rewiring techniques, it is **often comparable** to specialized heterophilous GNNs, with a standout improvement on **Roman-empire** but with no major gains elsewhere. Additionally, the absence of a **DHGR** comparison weakens conclusions about superiority among heterophily-oriented rewiring methods. Including DHGR would strengthen the empirical case.

4. **When to prefer rewiring over heterophilous GNNs.**
I am unsure as to when a practitioner should choose heterophilous GNNs or REFine. Are there any benefits of using the rewiring technique over GNNs other than just performance? If the performances are similar, why should we want to use a rewiring method over a heterophilous GNN? The paper should clarify when to choose rewiring (e.g., datasets where feature/label-driven reference graphs are reliably more homophilous), and provide guidelines on when specialized heterophilous GNNs remain the better choice.

**Questions:**

My questions largely stem from the limitations:

1. What are the advantages and disadvantages of REFine in comparison to DHGR? What are the conceptual differences given DHGR also constructs a rewired graph based on node features similarities and train labels?
2. How do the theoretical results meaningfully build on existing spectral analyses relating smoothness/energy to homophily/heterophily?
3. Given the connection between oversmoothing, oversquashing, and heterophily, what are the connections between existing rewiring approaches that address oversmoothing and oversquashing to heterophily? why don't their constructions mitigate the heterophily problem?
4. Why is DHGR not a baseline? How does DHGR perform in comparison to REFine?
5. It would be good as to get an intuition for why we would consider using a rewiring approach over a heterophilous GNN. The heterophilous GNNs perform closely in many cases, and if we have established heterophilous GNNs, when is rewiring a better alternative?

---

> ### Author Response · Authors · 2025-12-01
> **Response to reviewer M5SM - part 1**
>
> Thank you for your helpful feedback. We appreciate the time you took to review our work and provide valuable suggestions.
>
> ### Positioning vs. prior rewiring for heterophily
>
> Thank you for the insightful comment. We agree that DHGR is an important reference point, and we clarify below how REFine differs conceptually, methodologically, and in scope.
>
> First, while both REFine and DHGR use labels and features to guide rewiring, REFine proposes a general and theoretically grounded framework for homophily-improving rewiring, whereas DHGR is a specific similarity-learning model optimized for heterophilic graphs. Our framework explicitly analyzes why increasing homophily can improve message passing (Theorem 1), and we study when and why homophily-based rewiring succeeds or fails (e.g., the generalization issues that arise when using only training labels). This explanatory perspective is absent in DHGR.
>
> Second, REFine is model-agnostic and modular: any reference graph can be plugged into our pipeline. The “label-driven diffusion” graph we use in experiments is just one instance. In fact, DHGR itself can be viewed as another instance of our framework - one could simply use the DHGR similarity metric as the reference graph inside REFine. Our contribution is therefore not the specific reference graph construction, but the rewiring principle and its accompanying theoretical and empirical analysis.
>
> Third, REFine is substantially simpler and easier to deploy. DHGR learns a similarity matrix using a neural module and relies on distributions of multi-hop neighborhoods, masking rules, similarity learning, and scalable implementation tricks. In contrast, REFine does not require learning a parametric adjacency, which keeps complexity low and avoids the heavier machinery characteristic of GSL systems. This simplicity is one of our explicit design goals.
>
> Finally, while we agree that empirical comparisons to DHGR would be informative, DHGR is a GSL-style, learnable graph-optimization method, much more complex, and not immediately comparable or necessarily fair to evaluate against a non-learnable rewiring approach. Moreover, it is not standard practice for rewiring papers to include comparisons to GSL methods. At the time we conducted our experiments, an official implementation of DHGR was not publicly available, which prevented us from including a direct empirical comparison. Now that an official implementation of DHGR is publicly available, we have run their code under our evaluation protocol and report the results in the table below. As shown, REFine matches or outperforms DHGR on most datasets, and we will include these results in the revised version of the paper.
>
> |    Dataset   |   GCN+DHGR   |   GCN+REFine  |   GAT+DHGR   |   GAT+REFine  |
> |:------------:|:------------:|:-------------:|:------------:|:-------------:|
> |   Wisconsin  |   80.6±1.8   |  **82.5±1.6** |   81.7±1.4   |  **84.9±1.3** |
> |    Cornell   |   67.8±2.1   |  **71.3±1.5** | **75.1±1.4** |      74±2     |
> |     Texas    |   72.7±1.9   |  **79.1±1.6** |   70.2±2.4   |  **82.4±2.1** |
> |   Chameleon  |   41.1±1.1   |  **44.1±1.1** |   41.8±0.6   |  **43.5±1.8** |
> |   Squirrel   |   39.1±0.3   |  **41.1±0.7** |   37.6±0.5   |  **38.8±0.5** |
> |  BlogCatalog |   78.3±0.5   |  **85.2±0.3** |   83.2±0.6   |  **85.9±1.3** |
> |   Tolokers   |   77.2±0.3   |   **78±0.2**  |   79.2±0.2   |  **79.7±0.3** |
> |   Questions  |   66.9±1.4   |  **70.3±0.4** |      OOM     |  **66.6±0.5** |
> | Roman-empire |   56.1±0.2   |  **58.8±0.2** |   27.5±0.8   |  **28.5±0.7** |
> |     Actor    | **31.4±0.3** |    31.3±0.4   |   32.8±0.3   |  **35.1±0.3** |
> |      BGP     |   57.3±0.4   |  **59.3±0.2** |   63.2±0.4   |  **63.3±0.3** |
> |    Genius    |      OOM     | **83.8±0.04** |      OOM     | **83.6±0.03** |
>
> In summary, REFine provides (i) a principled rewiring framework with theoretical justification, (ii) a unifying view under which DHGR can be seen as a special case, and (iii) new insights on homophily-performance tradeoffs, beyond simply proposing another label-guided similarity function.

---

> ### Author Response · Authors · 2025-12-01
> **Response to reviewer M5SM - part 2**
>
> ### Incremental theory relative to prior spectral/smoothness analyses
>
> Thank you for pointing this out. Our analysis is indeed related in spirit to prior spectral/smoothness results such as Thm.3 in Zhu et al. (2020), but it plays a different role and goes beyond merely restating that heterophilic signals are “high frequency”. Thm.3 in Zhu et al. compares the spectral energy of label vectors themselves under different label-homophily levels. In contrast, our Thm.1 is formulated for learned node embeddings that are linearly separable by some classifier, and establishes an explicit quantitative lower bound on their Dirichlet energy that depends explicitly on the edge homophily. This makes the tension between (i) the smoothing bias of message-passing GNNs and (ii) the requirement to obtain linearly separable embeddings explicit: as homophily decreases, the smoothing effect of message passing is more likely to violate the separation inequality, causing the embeddings to become no longer linearly separable. This directly motivates increasing edge homophily via rewiring, rather than relying solely on architectural modifications.
>
> Building on this, Prop.1--2 together with our magnitude-of-improvement and concentration bounds characterize, for the first time to our knowledge, when reference-graph--guided edge addition or deletion provably increases homophily and by how much, in expectation.
>
> We then validate and refine these results with controlled simulations on real graphs using synthetic reference graphs of varying homophily (Sec.4.2): we systematically vary the homophily of the reference graph and the number of added/deleted edges $k$, and observe (i) regimes where increasing homophily improves accuracy, (ii) regimes where homophily increases but accuracy degrades due to over-squashing, and (iii) failure cases when the theoretical conditions are violated. These experiments reveal non-trivial limitations of the idea that “more homophily is always better” and provide concrete guidance on when and how REFine should be applied.
>
> We will clarify these distinctions with respect to Zhu et al. and more clearly emphasize that our theory is tailored to homophily-enhancing rewiring.
>
> ### Missing critical baseline and mixed empirical gains
>
> Thank you for the thoughtful feedback. We would like to clarify the empirical positioning of our method and why the results meaningfully support our claims.
>
> First, the goal of REFine is not to outperform every specialized heterophilous GNN, but to show that simple, standard message-passing GNNs, when combined with theoretically grounded homophily-enhancing rewiring, can match or exceed the performance of much more complex architectures. This is precisely what Table 2 demonstrates: on the majority of datasets, GCN/GAT/APPNP + REFine reach the performance level of specialized heterophily GNNs without any architectural modifications. The Roman-empire dataset shows especially large gains, but strong improvements also appear on Wisconsin and Chameleon. The key takeaway is not only the magnitude of improvement, but the fact that rewiring alone closes the gap to heterophily-oriented models, which is a central message of the paper.
>
> Second, regarding DHGR, as noted earlier, DHGR is a GSL-style, learnable graph-optimization method, and rewiring papers do not typically compare against such GSL frameworks. Nonetheless, we agree that a comparison is informative; we have now run the official DHGR implementation under our evaluation protocol (results presented above) and will include these results in the revised version.
>
> We will make these points clearer and include additional baselines where feasible in the revised version.

---

> ### Author Response · Authors · 2025-12-01
> **Response to reviewer M5SM - part 3**
>
> ### When to prefer rewiring over heterophilous GNNs
>
> Thank you for raising this important point. We agree that practitioners need concrete guidance on what to do when they first receive a new graph.
>
> In practice, we propose the following simple procedure (detailed in Appendix C.3). Given a new dataset, we (1) build a small sampled subgraph using training nodes and held-out validation nodes (whose labels are not used to construct the reference graph), (2) compute the empirical homophily of the original graph on this sample, and (3) compute the empirical homophily of the reference graph constructed by REFine on the same sample. If the reference graph is clearly more homophilous than the original graph while preserving enough edges per node, then our theoretical results predict that rewiring is likely to improve performance, and using REFine with a standard GNN is a good choice.
>
> Conversely, if the sampled reference graph is not more homophilous than the original graph, or if constructing a meaningful reference graph is difficult because features and labels are weakly aligned, then neither REFine nor most specialized heterophilous GNNs, which typically rely on similar assumptions, are expected to provide a clear advantage; indeed, in our experiments we did not observe systematic gains from specialized architectures. Their main practical benefit is that they avoid the graph preprocessing required by rewiring, so in scenarios where one wishes to avoid any preprocessing step, a specialized heterophilous GNN may still be preferable.
>
> ## Questions
> ### Advantages and disadvantages of REFine in comparison to DHGR
>
> Thank you for the question. While both REFine and DHGR utilize features and labels to guide rewiring, they differ fundamentally in purpose, design, and complexity. REFine provides a general, theoretically grounded framework for homophily-improving rewiring, whereas DHGR is a specific GSL-style, learnable similarity-optimization method tailored to heterophilic graphs. Our method aims to explain when and why increasing homophily helps (Theorem 1, Propositions 1–2) and provides conditions under which rewiring improves or harms performance, an analysis not present in DHGR.
>
> Conceptually, REFine is model-agnostic, modular, and non-learnable: any reference graph (including one derived from DHGR’s similarity metric) can be plugged into our pipeline. DHGR, in contrast, relies on a learned similarity matrix, autoencoder-based class embeddings, and multi-hop neighborhood encoding, making it substantially more complex and computationally heavier.
>
> In practice, REFine is lightweight and architecture-independent, whereas DHGR introduces significant training overhead. As shown in the table presented above, REFine also outperforms DHGR on most datasets. We will make these distinctions and the empirical comparison clearer in the revised version.
>
> ### Theoretical contribution beyond prior spectral analyses
>
> Our theoretical results build on prior smoothness–homophily analyses but go beyond them in two key ways. First, instead of analyzing the label vector or other fixed signals, we derive an explicit lower bound on the Dirichlet energy of learned node embeddings that remain linearly separable, showing that as graph homophily decreases, these embeddings must become increasingly non-smooth. This makes the tension between message-passing smoothing and the need for separability explicit, providing a concrete quantitative motivation for increasing homophily through rewiring rather than relying solely on architectural changes.
>
> Second, we analyze the effect of the rewiring operation itself: our propositions and bounds characterize when reference-graph–guided edge addition or deletion will provably increase homophily and by how much, in expectation. Prior analyses do not study how one graph can be used to modify another or when such modifications help or hurt. Our controlled simulations further validate these predictions and reveal regimes where homophily increases yet accuracy may still degrade, offering practical guidance beyond what earlier theory provides.

---

> ### Author Response · Authors · 2025-12-01
> **Response to reviewer M5SM - part 4**
>
> ### Why oversmoothing/oversquashing rewiring does not fix heterophily
>
> Existing rewiring methods targeting oversmoothing or oversquashing (e.g., curvature-based) are not designed to correct heterophily because they focus on improving information flow rather than correcting the label-mixing structure of the graph. These approaches typically add long-range or high-curvature edges to reduce bottlenecks or expand receptive fields, but they do not ensure that the added edges connect nodes of the same class - or that the resulting graph becomes more homophilous. As a result, in heterophilic settings they may actually reinforce incorrect message passing, since the added edges can amplify heterophilous mixing rather than counteract it.
>
> In contrast, REFine explicitly targets the label-alignment structure of the graph by using a homophilous reference graph to guide addition or deletion. This directly improves the compatibility between message passing and the class structure, which oversmoothing/oversquashing-focused rewiring methods do not guarantee.
>
> ### DHGR baseline
>
> As discussed above, DHGR is a GSL-style, learnable graph-optimization method-much more complex and not directly comparable to lightweight, non-learnable rewiring methods like REFine. At the time we conducted our experiments, an official implementation of DHGR was not publicly available, which prevented us from including a direct empirical comparison. Now that their code is available, we have run the official DHGR implementation under our evaluation protocol and will add the resulting comparison to REFine in the revised version.
>
> ### When is rewiring preferable to heterophilous GNNs?
>
> Rewiring provides a structurally different way to address graph-related issues: instead of designing a specialized architecture, it modifies the graph so that standard message passing becomes more effective. This mirrors how oversquashing can be alleviated either through architectural changes or through rewiring; heterophily can be approached in the same spirit.
>
> In our case, when the reference graph is reliably more homophilous, REFine consistently improves accuracy and often lifts simple GNNs to the level of specialized heterophilous models. In these regimes, rewiring is typically more efficient, architecture-agnostic, and avoids introducing extra learned parameters. While specialized heterophilous GNNs may still be chosen in settings where such homophily gains cannot be obtained (or when one wishes to avoid a preprocessing step), rewiring provides a lightweight and broadly applicable alternative whenever a more homophilous reference graph is available.

---

### Official Review · Reviewer_prfN · 2025-10-30

**Soundness:** 3
**Presentation:** 2
**Contribution:** 2
**Rating:** 4
**Confidence:** 3

**Summary:**

The paper addresses the challenge of poor GNN performance on heterophilic graphs. It explores graph rewiring as a strategy to improve GNN performance with theoretical and empirical study. It specifically introduces a label-driven diffusion method to construct the reference graph from node features and available labels, to rewire graphs for increasing homophily. Extensive experiments could show the effectiveness of the proposed method.

**Strengths:**

1. The idea of addressing the homophily–heterophily issue through graph rewiring is interesting and provides a different perspective compared to traditional architectural modifications.

2. The paper combines theoretical analysis regarding the reference graph with a comprehensive empirical study, and the explanations are generally clear and well-motivated.

3. Some experimental results indeed show significant performance improvements over certain baselines.

**Weaknesses:**

1. The proposed method lacks clear novelty. Label-guided graph rewiring has already been studied, for example, in *Bose, K., Banerjee, S., & Das, S. (2025). “Can Graph Neural Networks Tackle Heterophily? Yes, With a Label-Guided Graph Rewiring Approach!” IEEE TNNLS*, and the core technical component "label-driven diffusion" used here appears directly derived from existing work Mendelman
& Talmon (2025)., which substantially weakens the contribution.

2. The paper does not convincingly justify why increasing graph homophily is the right direction. Ideally, models should handle both homophilic and heterophilic structures without modifying the graph itself. More puzzlingly, Table 2 shows that the proposed method, which increases homophily, performs better even on heterophilic GNNs (e.g., H2GCN) — this seems conceptually inconsistent and requires further discussion.

3. The baseline comparison is outdated and incomplete. The latest strong methods such as *Bose et al., 2025 (label-guided rewiring) and Barbero et al., 2023, “Locality-Aware Graph Rewiring in GNNs,” ICLR 2024*, and above TNNLS 2025 work are not included.

4. The organization and presentation need improvement. Many experimental results that belong in the main text are buried in the appendix, with duplicated tables (E.1 COMPLETE RESULTS seems same as Table.1), and no bottomline in many tables. The paper feels somewhat rushed.

**Questions:**

See weakness.

---

> ### Author Response · Authors · 2025-12-01
> **Response to reviewer prfN - part 1**
>
> Thank you for your helpful feedback. We appreciate the time you took to review our work and provide valuable suggestions.
>
> ### Novelty
>
> Thank you for the comment. While several rewiring methods use labels in some form, our work is the first to introduce a systematic and theoretically grounded framework for improving homophily through rewiring, with explicit guarantees on when addition or deletion increases homophily. This framework is general and agnostic to the specific construction of the reference graph - as stated in Section 5, label-driven diffusion is only one possible instantiation of the reference graph within the broader framework, not the core contribution itself. Beyond this, our work contributes several insights that were not available in prior label-guided rewiring methods: we analyze why naive label-only rewiring fails, identify conditions under which improving homophily helps, and highlight regimes where homophily can increase while performance still degrades due to issues such as over-smoothing or over-squashing. These analyses expose fundamental trade-offs that previous work has not examined.
>
> Regarding the paper by Bose, Banerjee, and Das (2025), their method (LGR) was published very recently and developed in parallel to ours. Although both methods use labels, LGR is fundamentally different: it combines predicted class probabilities, autoencoder-based class embeddings, and a learnable similarity matrix, followed by a two-stage, learnable rewiring procedure. In contrast, our method is far simpler, non-learnable, and guided by a clean theoretical principle: rewiring according to a reference graph that satisfies homophily-improvement conditions guarantees expected improvement without learning an adjacency matrix or relying on heavy machinery such as autoencoders. We will clarify these distinctions more explicitly in the revised version.
>
> ### Increasing homophily
>
> Thank you for the comment. Our motivation for increasing homophily is formally established in Theorem 1, where we show that higher homophily leads to smoother yet still class-separable node embeddings. This directly benefits standard message-passing GNNs, whose core operation is a smoothing of neighboring representations. Thus, increasing homophily strengthens the very mechanism through which classical GNNs learn.
>
> Regarding architectures designed for heterophilic graphs, we agree that some models can better handle heterophily by construction. However, the reviewer’s interpretation of Table 2 is not accurate. We do not claim that our rewiring improves heterophilic-specialized GNNs such as H2GCN. Rather, Table 2 demonstrates a different point: standard GNNs (which naturally benefit from higher homophily) combined with our rewiring can match or outperform specialized heterophilic GNNs without rewiring. In other words, homophily-enhancing rewiring can elevate simple message-passing architectures to the level of more complex models, not that it improves the specialized ones themselves.
>
> We will clarify this distinction in the revised version of the paper.

---

> ### Author Response · Authors · 2025-12-01
> **Response to reviewer prfN - part 2**
>
> ### Baseline comparison
>
> Thank you for the suggestion. The method of Bose et al. (2025) was developed and published in parallel to our work, so we were not aware of it at submission time and could not include it as a baseline. Their method is also substantially more complex, requiring a learnable adjacency matrix, autoencoder-based class embeddings, and class-probability prediction modules, which makes comparison non-trivial and less aligned with our goal of a simple, non-learnable rewiring. We have attempted to reproduce their experiments following the publication of their code, but we encountered technical issues and were not yet able to obtain stable results. We will continue working on this and aim to include a comparison in the revised version.
> To address the reviewer’s concern regarding more recent baselines, we have now included results for AFRC (“Mitigating Over-Smoothing and Over-Squashing using Augmentations of Forman-Ricci Curvature”, LoG 2024), which is a strong and recent rewiring approach that extends BORF. Results are shown below (excluding earlier baselines for brevity):
>
> |    Dataset   | GCN+AFRC |   GCN+REFine  | GAT+AFRC |   GAT+REFine  | APPNP+AFRC |  APPNP+REFine |
> |:------------:|:--------:|:-------------:|:--------:|:-------------:|:----------:|:-------------:|
> |   Wisconsin  | 52.7±1.7 |  **82.5±1.6** | 53.3±2.2 |  **84.9±1.3** |  60.6±1.4  |   **86±1.4**  |
> |    Cornell   | 48.4±2.7 |  **71.3±1.5** | 48.1±2.1 |    **74±2**   |  51.6±1.6  |  **74.6±1.5** |
> |     Texas    | 57.3±2.3 |  **79.1±1.6** | 57.6±2.5 |  **82.4±2.1** |  63.2±1.8  |  **82.4±1.7** |
> |   Chameleon  |  41.6±1  |  **44.1±1.1** | 42.2±1.2 |  **43.5±1.8** |  40.4±0.9  |  **44.5±1.2** |
> |   Squirrel   | 40.2±0.5 |  **41.1±0.7** | 36.1±0.6 |  **38.8±0.5** |   36±0.5   |  **38.8±0.8** |
> |  BlogCatalog | 77.2±0.5 |  **85.2±0.3** | 82.5±1.5 |  **85.9±1.3** |  95.5±0.3  |  **95.7±0.2** |
> | Roman-empire | 40.5±0.3 |  **58.8±0.2** | 18.2±0.5 |  **28.5±0.7** |  20.1±0.1  |  **30.8±0.5** |
> |     Actor    | 28.5±0.3 |  **31.3±0.4** | 29.2±0.3 |  **35.1±0.3** |  33.3±0.2  |  **34.8±0.3** |
> |      BGP     | 52.4±0.5 |  **59.3±0.2** |  63±0.3  |  **63.3±0.3** |  63.1±0.4  |  **64.3±0.3** |
> |    Genius    |    T/O   | **83.8±0.04** |    T/O   | **83.6±0.03** |     T/O    | **83.6±0.04** |
>
> As seen, our method matches or outperforms AFRC in most cases. We will incorporate these results into Table 1 in the revised version.
>
> We also include DHGR as a new baseline, see the response to reviewer M5SM.
>
> ### Presentation
>
> Thank you for the valuable feedback. We apologize for the confusion regarding the tables. Table E.1 in the appendix is not a duplicate of Table 1: it reports the same mean accuracies but additionally includes the Standard Error of the Mean, which we could not fit into the main paper due to space limitations.
>
> Regarding the presentation, both Table 1 and Table 2 do include bottom-line takeaways, which are echoed in the main text through statements such as: “Our REFine outperforms the baseline methods in most cases, significantly improving performance compared to the leading baseline”. In the revised version, we will make these conclusions more visually explicit within the tables themselves to avoid any ambiguity.
>
> We take the reviewer’s concern about the organization very seriously. In the revised version, the page limit increases by one page, which will allow us to move several key results from the appendix into the main text, reduce redundancy, and improve the flow. We will use this additional space to reorganize the experiments section, streamline the tables, and improve the overall clarity and readability of the paper.

---

### Official Review · Reviewer_SpJs · 2025-11-02

**Soundness:** 2
**Presentation:** 3
**Contribution:** 2
**Rating:** 4
**Confidence:** 5

**Summary:**

This paper introduces REFine, a graph rewiring framework designed to enhance graph homophily using a reference graph, thereby improving the performance of standard Graph Neural Networks (GNNs) on heterophilic graphs. The authors provide a theoretical foundation linking edge homophily, embedding smoothness, and GNN performance, which motivates homophily enhancement. They propose a principled rewiring method guided by a reference graph, with theoretical guarantees on homophily improvement under certain conditions. A label-driven diffusion process is introduced to construct a homophilic reference graph from node features and training labels.

**Strengths:**

S1 The paper establishes a clear theoretical connection between graph homophily and the smoothness of GNN embeddings (Theorem 1), providing a strong motivation for homophily-enhancing rewiring.
S2 The experimental section is thorough, evaluating the method across a diverse set of 13 datasets with varying sizes and homophily levels.
S3 The use of the METIS algorithm for graph partitioning, coupled with parallelizable per-cluster computations, makes the method highly scalable to large graphs.

**Weaknesses:**

W1 The effectiveness of the reference graph construction hinges on the assumption that node feature similarity is indicative of label similarity. In domains where this assumption does not hold, the method's performance may be limited. While the label-driven diffusion aims to mitigate this, the fundamental dependency remains a potential limitation.
W2 The framework is specifically designed and evaluated for the node classification task. Its applicability to other fundamental graph learning tasks, such as graph classification or link prediction, is not explored or discussed, which may limit its perceived utility for the broader graph learning community.
W3 The method involves several key hyperparameters, including the kernel scale ε, the choice between edge addition/deletion, and the number of edges k to rewire. While a grid search is employed, the performance is contingent on proper tuning, which could be computationally expensive and less user-friendly in practice.

**Questions:**

Q1 The current framework performs either edge addition or deletion in a single rewiring step. Have the authors considered a strategy that performs both operations simultaneously, which could offer finer control over the resulting graph topology?
Q2 The reference graph is constructed in a fixed, non-learnable manner. Could the performance be further improved by integrating the reference graph construction into an end-to-end, trainable framework, perhaps drawing inspiration from GSL paradigms in future work?
Q3 How does the method perform on graphs that are extremely sparse (leading to potential disconnection) or excessively dense (where rewiring might have minimal relative impact)? Are there specific regimes where the method is less effective?

---

> ### Author Response · Authors · 2025-12-01
> **Response to reviewer SpJs - part 1**
>
> ## Part 1
>
> Thank you for your helpful feedback. We appreciate the time you took to review our work and provide valuable suggestions.
>
> ### Node Feature Similarity
>
> Thank you for pointing this out, we agree that our approach fundamentally relies on the assumption that feature similarity is at least partially aligned with label similarity. We explicitly state this assumption in the paper, and label-driven diffusion is intended to make the reference graph less sensitive to local feature noise and limited labels, but it cannot fully overcome a complete feature–label mismatch. In such domains, we expect REFine to be less effective, and we view this as an important limitation and a motivation for future work on alternative reference graph constructions that rely more heavily on structure, labels, or domain-specific similarity measures rather than raw features alone.
>
> ### Other tasks (graph-level, link prediction)
>
> Thank you for the thoughtful comment. You are right that the framework is specifically designed and evaluated for the node classification task. In fact, we explicitly acknowledge this limitation in the Conclusion, where we state that our framework, similar to other homophily-enhancing approaches, is currently inapplicable to graph classification. We will clarify this point earlier in the paper to avoid confusion.
>
> We appreciate the reviewer’s suggestion regarding broader applicability. Indeed, most rewiring methods in the literature, including recent works [1, 2], primarily focus on node-level or graph-level prediction tasks. As noted in the paper, our method is better suited to non-graph-level tasks due to its reliance on node-wise homophily and message-passing dynamics.
>
> That said, we find the reviewer’s suggestion to explore link prediction particularly valuable. Since our framework explicitly ties homophily enhancement to improved message passing, and given recent evidence that increased homophily can benefit link prediction performance [3], extending REFine to link prediction is a promising direction for future work.
>
> ### Hyperparameters
>
> Thank you for the comment. While our method involves several hyperparameters, this is standard for rewiring approaches - e.g., SDRF requires tuning continuous parameters like temperature and removal bounds, as well as the number of iterations.
>
> While we tune $\epsilon$ in our experiments, using a Gaussian kernel is standard practice, and $\epsilon$ can be set via common practices, such as the median of squared distances. In the table below, we compare the downstream accuracy using our tuned $\epsilon$ with the median-based alternative, alongside the original graph baseline:
>
> | Dataset     | Tuned scale | Median scale | Original graph accuracy | Tuned scale accuracy | Median scale accuracy |
> |-------------|:-----------:|:------------:|:-----------------------:|:--------------------:|:---------------------:|
> | Wisconsin   |     100     |      125     |         57.2±1.9        |       82.5±1.6       |        82.4±1.3       |
> | Cornell     |     100     |      128     |         51.8±1.3        |       71.3±1.5       |        71.6±1.4       |
> | Texas       |     100     |      104     |         59.7±2.6        |       79.1±1.6       |        78.5±1.6       |
> | Chameleon   |     100     |      17      |         41.3±0.6        |       44.1±1.1       |        42.6±1.5       |
> | Squirrel    |      10     |      37      |         40.7±0.4        |       41.1±0.7       |         41±0.5        |
> | BlogCatalog |      10     |      114     |         77.6±0.8        |       85.2±0.3       |        82.6±0.8       |
>
>
> The median-based $\epsilon$ often matches the tuned value and yields similar performance.
> Overall, the hyperparameter space is small - feature vs. feature+label (2 options), add vs. remove (2 options), number of edges, and optionally $\epsilon$, which can be set using a standard rule like the median. Please note that this is comparable to or even simpler than other rewiring methods.
>
> References:
>
> [1] Revisiting Over-smoothing and Over-squashing Using Ollivier-Ricci Curvature (ICLR 2023)
>
> [2] Mitigating Over-Smoothing and Over-Squashing using Augmentations of Forman-Ricci Curvature (LoG 2024)
>
> [3] On the Impact of Feature Heterophily on Link Prediction with Graph Neural Networks (NeurIPS 2024)

---

> > ### Author Response · Authors · 2025-12-01
> > **Response to reviewer SpJs - part 2**
> >
> > ## Questions
> > ### Simultaneous edge addition and deletion
> >
> > Thank you for the insightful suggestion. Our current framework indeed performs either edge addition or deletion per rewiring step, but we explicitly mention simultaneous edge addition and deletion as a future direction in the Conclusion. We agree that such a strategy could, in principle, offer finer control over the resulting topology. However, it would also introduce additional tuning complexity, which may make the method less user-friendly. Moreover, we have found a correlation between the original graph connectivity and the effectiveness of these operations on performance, when we measure connectivity by the average shortest path length normalized by the number of nodes in the graph: when the graph is more connected, edge deletion is more effective than edge addition, and when the graph is less connected, edge addition is more beneficial. For these reasons, we focused on the simpler, single-operation setting in this work and leave a joint add+delete variant for future research.
> >
> > ### Learnable reference graph
> >
> > Thank you for the suggestion. Integrating the reference graph construction into an end-to-end, learnable GSL-style framework is indeed an interesting direction and could potentially improve performance, though it would also make the method significantly more complex and harder to tune. We note that our framework is agnostic to how the reference graph is constructed. As stated in the Proposed Method section, any reference graph that satisfies the homophily-improvement conditions can be used.
> >
> > ### Sparse/dense graphs
> >
> > Thank you for the question. As mentioned before, we observe a correlation between the graph’s connectivity, measured by the average shortest path length normalized by the number of nodes, and which operation is more effective. In highly connected graphs, edge deletion tends to be more beneficial than edge addition, since there is enough redundancy in the connectivity and removing edges can improve homophily without harming information flow. In weakly connected graphs, edge addition is typically preferable, as further deletion can increase the risk of disconnection and exacerbate over-squashing.

---

### Author Response · Authors · 2025-12-01

We thank all reviewers for the time and effort invested in evaluating our submission. Due to the unusual circumstances, the discussion phase was terminated early, so we use this general response to summarize the reviews and how we addressed their main concerns.

Reviewer SpJs: The reviewer raised concerns about reliance on feature–label alignment, the focus on node classification, and tuning a few hyperparameters. We clarified that REFine explicitly assumes at least partial feature–label alignment and naturally degrades when this fails (a stated limitation and avenue for future work), that the framework is intentionally tailored to node classification, and that the hyperparameter space is small and comparable to or simpler than other rewiring methods, with standard heuristics working well in practice. We believe this fully addresses the reviewer’s concerns.

Reviewer prfN: The reviewer questioned novelty over label-guided rewiring, the motivation for increasing homophily, and baseline completeness. We emphasized that our main contribution is a general, theoretically grounded rewiring framework with provable homophily-improvement guarantees and an analysis of when homophily helps or hurts, rather than a particular label-guided similarity. We distinguished REFine from recent label-guided methods that learn an adjacency matrix and are substantially more complex, whereas our approach is non-learnable and principle-driven. We also clarified that we do not claim homophily is universally necessary, only beneficial in the practically relevant regimes we study, added AFRC as a recent baseline, and will include further comparisons where feasible.

Reviewer M5SM: This reviewer’s main concerns centered on positioning versus DHGR. We clarified that DHGR is a specific, learnable GSL-style method, whereas REFine is a general, modular framework into which DHGR can be seen as a special case via its similarity metric, and that our contribution lies in the rewiring principle and its theoretical analysis rather than any particular similarity construction. We further explained that our theory goes beyond prior smoothness analyses by linking linearly separable embeddings and Dirichlet energy as a function of homophily and by analyzing when reference-graph–guided edge addition/deletion provably improves homophily. Using the official DHGR implementation under our protocol, we added direct comparisons showing REFine outperforms DHGR on most datasets.

Reviewer CvPR: The reviewer raised a potential mismatch between global theory and cluster-based implementation, questioned our focus on homophily, and pointed to practical concerns about memory and clustering. We clarified that, under mild assumptions, our theoretical results extend naturally to the clusterwise setting because global homophily is a weighted combination of cluster-wise homophily, so improving each cluster improves the overall quantity. We positioned our work as complementary to “homophily is not necessary” results, which assume highly informative label distributions, whereas real heterophilic benchmarks often have low label informativeness, making homophily enhancement useful in practice. We also clarified that the reference graph is only used during offline rewiring (no doubled memory during GNN training), that cluster-size sensitivity is limited, and that we will improve presentation with summary rows and clearer cross-dataset references.

Reviewer PKvY: This review is largely based on misunderstandings of prior work and our contributions. We clarified that recent works on GNNs under heterophily do not claim that increasing homophily cannot help, but instead show that plain GNNs can succeed in special cases with highly informative local label distributions - conditions rarely met on real heterophilic benchmarks, as indicated by low “label informativeness” in "Characterizing Graph Datasets for Node Classification: Homophily–Heterophily Dichotomy and Beyond" (NeurIPS 2023); our theory focuses on the message-passing bias whereby decreasing homophily forces linearly separable embeddings to become highly non-smooth and vulnerable to standard smoothing, motivating homophily-enhancing rewiring in realistic regimes. We corrected the misinterpretation of Theorem 1, emphasizing that it assumes standard class-wise linear separability and quantifies its cost on a given graph, with relevance to GNNs arising from combining this bound with the smoothing bias of message-passing architectures. Finally, we clarified that we use corrected splits where appropriate, adopt a unified modest-capacity protocol across all models (standard in rewiring work), and focus on relative gains between each backbone and its REFine-augmented counterpart; thus the reviewer’s concerns about absolute accuracies and dataset choices do not indicate fundamental flaws in our approach.

---

### Meta-Review · Area_Chair_28iQ · 2026-01-06

**Summary:**

Reviewers’ main concerns centered on (i) novelty, questioning overlap with prior label-guided rewiring and GSL methods; (ii) conceptual justification, asking why increasing homophily is the right direction; (iii) assumptions, particularly reliance on feature–label alignment in constructing the reference graph; (iv) scope, noting the method is limited to node classification; and (v) evaluation completeness, including missing recent baselines and practical issues such as hyperparameter tuning and scalability.

The authors responded by clarifying that the core contribution is a general, theoretically grounded rewiring framework with provable homophily-improvement conditions, rather than a specific label-guided similarity; grounding homophily enhancement via a formal link between homophily, smoothness, and linear separability; explicitly acknowledging feature–label alignment and task scope as limitations; adding newer baselines and direct comparisons (e.g., DHGR/AFRC); and justifying clustering, scalability, and modest hyperparameter sensitivity.

Overall, most concerns were addressed, but the novelty issue remains. Given the current ratings and feedback from the reviewers, this paper has not met the bar for ICLR 2026.

**Reviewer Concerns:**

Most concerns were addressed, but the novelty issue remains

**Reviewer Scores:**

Reviewer PKvY would probably raise their score to 4; Reviewer M5SM would probably raise their score to 4 as well. The other reviewers may maintain the original scores.

---

### Decision · Program_Chairs · 2026-01-26

Reject